# Stability and Generalization of Nonconvex Optimization with Heavy-Tailed Noise

Hongxu Chen [1 2]   Ke Wei [1]   Xiaoming Yuan [2]   Luo Luo [1]

## Abstract

The empirical evidence indicates that stochastic optimization with heavy-tailed gradient noise is more appropriate to characterize the training of machine learning models than that with standard bounded gradient variance noise. Most existing works on this phenomenon focus on the convergence of optimization errors, while the analysis for generalization bounds under the heavy-tailed gradient noise remains limited. In this paper, we develop a general framework for establishing generalization bounds under heavy-tailed noise. Specifically, we introduce a truncation argument to achieve the generalization error bound based on the algorithmic stability under the assumption of bounded $p$th centered moment with $p \in (1, 2]$. Building on this framework, we further provide the stability and generalization analysis for several popular stochastic algorithms under heavy-tailed noise, including clipped and normalized stochastic gradient descent, as well as their mini-batch and momentum variants.

## 1. Introduction

Over the past decades, machine learning develops very rapidly and has achieved remarkable progress in many areas including computer vision (Voulodimos et al., 2018; Szeliski, 2022), natural language processing (Devlin et al., 2019; Chowdhary, 2020), and reinforcement learning (Sutton & Barto, 1998; Guo et al., 2025). In many machine learning applications, the task of training the model can be formulated as an optimization problem, where one seeks parameters that yield strong generalization performance. A canonical formulation is to find the parameter by minimizing

[1]School of Data Science, Fudan University, Shanghai, China [2]Department of Mathematics, The University of Hong Kong, Hong Kong SAR, China. Correspondence to: Luo Luo <luoluo@fudan.edu.cn>.

*Proceedings of the 43$^{rd}$ International Conference on Machine Learning*, Seoul, South Korea. PMLR 306, 2026. Copyright 2026 by the author(s).

the population risk as follows

$$\min_{x \in \mathbb{R}^d} F(x) := \mathbb{E}_{\xi \sim \mathcal{D}}\big[f(x; \xi)\big], \tag{1}$$

where $\xi$ denotes data that follows a distribution $\mathcal{D}$ and $f(x; \xi)$ is the corresponding loss function. In practice, the explicit expression of the objective in formulation (1) is typically unavailable, but can be approximated via a training dataset $S = \{\xi_1, \ldots, \xi_n\}$ of $n$ i.i.d. samples drawn from $\mathcal{D}$. This yields the empirical risk minimization problem

$$\min_{x \in \mathbb{R}^d} F_S(x) := \frac{1}{n} \sum_{i=1}^n f(x; \xi_i). \tag{2}$$

Since the training dataset often contains a large number of samples, it is prohibitively expensive to compute the full gradient of (2). Thus the stochastic optimization methods that exploit the finite-sum structure are natural alternatives. A representative one is stochastic gradient descent (SGD), which at each iteration uses the gradient of a single randomly sampled data point to approximate the full gradient. Its low per-iteration cost and simple update rule make it effective for training large models. The analysis of SGD can be traced back to Robbins & Monro (1951). For convex problems, convergence can be guaranteed by classical results (Polyak & Juditsky, 1992; Nemirovski et al., 2009). For nonconvex problems, analyses typically rely on the magnitude of gradients (Ghadimi & Lan, 2013; Bottou et al., 2018). Moreover, many variants of SGD, including momentum methods (Polyak, 1964; Nesterov, 1983; Sutskever et al., 2013; Liu et al., 2020), variance reduction (Johnson & Zhang, 2013; Zhang et al., 2013; Defazio et al., 2014; Agarwal & Bottou, 2015; Woodworth & Srebro, 2016; Schmidt et al., 2017; Allen-Zhu, 2018; Kovalev et al., 2020; Fang et al., 2018; Li et al., 2021a), and adaptive learning rates (Duchi et al., 2011; Kingma, 2014; Zeiler, 2012; Reddi et al., 2018; Loshchilov & Hutter, 2019; Ivgi et al., 2023), have improved theoretical bounds or empirical performance.

In real-world applications, we are more interested in the generalization measured on the population risk (1), while optimization algorithms focus on the empirical risk (2). We use notation $A(S)$ to represent the output of algorithm $A$ for solving problem (2) with training dataset $S$. For convex

objective, the generalization error is commonly measured by the difference of function value, i.e., $F(A(S)) - F_S(A(S))$. Specifically, Bousquet & Elisseeff (2002) introduced the notion of uniform stability and established an approach for the analysis of generalization bounds from this perspective. In subsequent work, Hardt et al. (2016) analyzed stability and generalization of SGD under the assumptions of Lipschitz and smoothness, providing theoretical explanation for the early stopping. The smoothness assumption is removed by Lei & Ying (2020). Later, Feldman & Vondrak (2019) strengthened the stability-based bounds from expectation to near-optimal high-probability guarantees. For nonconvex objective, the generalization error is usually measured by the gradient discrepancy, i.e., $\|\nabla F(A(S)) - \nabla F_S(A(S))\|$. Specifically, Lei (2023) introduced the notion of uniform stability with respect to the gradient, yielding a general framework for the generalization analysis for nonconvex stochastic optimization algorithms. In addition, several recent works analyzed the stability and generalization of problems with minimax formulations (Lei et al., 2021; Zhang et al., 2024) and distributed settings (Deng et al., 2024; Le Bars et al., 2024; Zhu et al., 2024; Zeng & Lei, 2025).

However, the existing results typically rely on the standard bounded-variance assumption. Recent empirical evidence suggests that, in the popular tasks such as deep neural network training (Simsekli et al., 2019; Zhang et al., 2020; Ahn et al., 2024; Battash et al., 2024) and reinforcement learning (Garg et al., 2021), the noise of stochastic gradients follows heavy-tailed distribution. Simsekli et al. (2019) modeled this phenomenon using the $\alpha$-stable distribution and analyzed the dynamics of SGD through a Lévy-driven SDE. In optimization theory, a common way to formalize the heavy-tailed noise is the assumption of bounded $p$th centered moment, i.e., there exist $p \in (1, 2]$ and $\sigma_p > 0$ such that for all $x \in \mathbb{R}^d$, it holds

$$\mathbb{E}_{\xi \sim \mathcal{D}} \left[ \|\nabla f(x; \xi) - \nabla F(x)\|^p \right] \leq \sigma_p^p. \qquad (p\text{-BCM})$$

Under the assumption of $p$-BCM, Zhang et al. (2020) established the in-expectation convergence guarantees for clipped SGD. Cutkosky & Mehta (2021) derived the high-probability convergence guarantees accordingly. Nguyen et al. (2023) and Liu et al. (2023) further improved the high-probability analysis and developed variance-reduced clipped SGD methods (Liu et al., 2023). The recent works show that normalized SGD also guarantees the convergence of stochastic nonconvex optimization under the $p$-BCM assumption. Specifically, Hübler et al. (2025) show that normalized SGD with momentum or large batch size can achieve the optimal convergence rates like the counterparts of clipped SGD. In addition, Sun et al. (2025b) combined clipping and normalization with momentum to establish the sharper convergence rates by assuming the objective has Lipschitz continuous Hessian. He et al. (2025) further analyzed

Table 1. Comparison of generalization errors for nonconvex problems measured by $\|\nabla F(A(S)) - \nabla F_S(A(S))\|$. Here $\epsilon$ denotes the algorithmic stability parameter. Our bound holds for $p \in (1, 2]$ and reduces to the result of Lei (2023) when $p = 2$.

| Reference | Bounds | Assumption |
|---|---|---|
| Mei et al. (2018) | $O\left(\sqrt{d \log(n)/n}\right)$ | sub-Gaussian |
| Lei (2023) | $O(\epsilon + n^{-\frac{1}{2}})$ | bounded variance |
| Theorem 4.2 | $O(\epsilon + n^{-\frac{p-1}{p}})$ | $p$-BCM |

the acceleration by the high-order smoothness. Fang et al. (2026) showed that step normalization remains robust for stochastically preconditioned SGD under heavy-tailed noise. Yu et al. (2026) established convergence guarantees for sign-based optimizers under a generalized heavy-tailed noise condition. Recently, distributed stochastic optimization under heavy-tailed noise has also been investigated (Yu et al., 2025; Sun et al., 2025a; Wang et al., 2026; Wu & Luo, 2026; Zhang & Gao, 2025).

The understanding of generalization analysis under the $p$-BCM assumption remains limited. One line of work by Raj et al. (2023a;b) established the algorithmic stability results for SGD under the general loss functions, while their analysis relies on continuous-time SDE arguments and Wasserstein distances. Recently, Sun et al. (2025c) analyzed the generalization of normalized SGD under the unbounded noise and a weak gradient Lipschitz condition, which is different from commonly used $p$-BCM assumption for heavy-tailed noise. Zeng & Lei (2026) established excess risk bounds for clipped SGD in the convex setting.

In this paper, we develop a systematic stability-based generalization analysis for stochastic nonconvex optimization under the $p$-BCM condition. We apply our framework to study two representative approaches for handling heavy-tailed noise, clipped SGD and normalized SGD, as well as their mini-batch and momentum variants. The main contributions of this paper are summarized as follows.

- We develop, to the best of our knowledge, the first connection between algorithmic stability and generalization error for nonconvex problems under the $p$-BCM condition. In particular, we show that the gap between population and empirical gradients can be bounded by $4\epsilon + C_p \sigma_p n^{-\frac{p-1}{p}}$, where $\epsilon$ is the algorithmic stability parameter and $C_p$ is a constant. This bound coincides with the result of Lei (2023) when $p = 2$. It is worth noting that our analysis is nontrivial, since we introduce a new technique of truncation argument to control heavy-tailed noise of the stochastic gradients. We compare our results with existing generalization bounds for nonconvex optimization problems in Table 1.

- We apply our stability-based generalization framework to analyze clipped and normalized SGD, as well as their mini-batch and momentum variants, yielding the risk bounds for nonconvex problems under heavy-tailed noise. For clipped SGD, we show that incorporating normalization and momentum does improve the stability and the generalization. We also analyze the mini-batch and the momentum variants of normalized SGD to achieve sharper population risk bounds than that of clipped SGD.

## 2. Related Work

**Stability and generalization.** The generalization gap between training and testing is a central topic in statistical learning theory. Two main approaches have been developed. The first is uniform convergence, which primarily depends on the hypothesis class and is typically algorithm-agnostic. Bartlett & Mendelson (2002) defined the Rademacher complexity and used capacity measures of function classes to bound the gap between empirical and population risks. Shalev-Shwartz et al. (2010) further discussed the relationship between stability and uniform convergence from the perspective of learnability. In the nonconvex setting, Mei et al. (2018) studied uniform convergence of gradients, characterizing function class complexity via covering numbers, and Foster et al. (2018) further derived uniform convergence bounds using Rademacher complexity. However, uniform convergence bounds often depend on the dimension, which makes them unfavorable for high-dimensional regimes commonly encountered in machine learning. The second line is based on algorithmic stability. The central idea is to quantify the effect of replacing a single training sample on the algorithm output and to obtain generalization bounds based on this sensitivity (Bousquet & Elisseeff, 2002). Hardt et al. (2016) established stability guarantees for stochastic gradient descent under Lipschitz and smoothness assumptions, which revealed the impact of the iterations on generalization and explained the role of early stopping. Moreover, Lei & Ying (2020) removed the smoothness assumption, and Feldman & Vondrak (2019) derived near-optimal high-probability bounds. For nonconvex problems, stability in gradients was established in Lei (2023). However, existing analyses typically require bounded variance. Establishing generalization guarantees under the $p$-BCM condition remains an open problem. Raj et al. (2023a;b) studied heavy-tailed noise modeled by $\alpha$-stable distributions and analyzed SGD through continuous-time SDEs. Recent work (Dang et al., 2025) extended this line of analysis to SGD with momentum.

**Clipped SGD.** Gradient clipping was introduced as a practical safeguard for exploding gradients in recurrent networks (Pascanu et al., 2013) and has since become a standard component of private learning (Abadi et al., 2016). In stochastic optimization, gradient clipping has often been studied under heavy-tailed gradient noise. For convex problems, clipped mirror-descent type methods and related techniques yield non-asymptotic guarantees, including high-probability results under weak tail assumptions (Nazin et al., 2019; Davis et al., 2021; Gorbunov et al., 2020). For nonconvex problems, Zhang et al. (2020) established convergence guarantees for clipped SGD and provided matching lower bounds, while Sadiev et al. (2023) and Nguyen et al. (2023) developed high-probability bounds for clipped stochastic methods under unbounded variance. Related high-probability analyses also combined clipping with momentum and normalization in nonconvex optimization (Cutkosky & Mehta, 2021; Liu et al., 2023). In addition, Mai & Johansson (2021) analyzed the stability and convergence of clipped SGD for non-smooth convex functions and Koloskova et al. (2023) established the convergence with a constant clipping threshold under bounded variance.

**Normalized SGD.** Gradient normalization traces back at least to Nesterov (2004), which analyzed the deterministic convex setting. Hazan et al. (2015) introduced normalized gradient descent in the stochastic setting and studied its convergence under quasi-convexity and local Lipschitz assumptions. Levy (2017) analyzed the convergence of adaptive normalized gradient descent for smooth objectives. In the nonconvex case, Cutkosky & Mehta (2020) combined momentum with normalized SGD and showed that momentum removed the requirement for large batches. Yang et al. (2023) established lower bounds for normalized SGD in stochastic setting and compared adaptive methods without the knowledge of problem specific parameters. Moreover, Hübler et al. (2024) derived nearly optimal rates for normalized SGD with momentum under relaxed smoothness. For heavy-tailed noise, Cutkosky & Mehta (2021) combined normalized SGD with gradient clipping and momentum, then provided a high-probability convergence analysis. Liu & Zhou (2025) revisited batched normalized SGD with momentum and pointed out that one can achieve the optimal rate without clipping. Hübler et al. (2025) further discussed limitations of clipping and established complexity bounds for normalized SGD with mini-batch or momentum, together with matching lower bounds. Sun et al. (2025b) studied normalized SGD with clipping and obtained sharper rates under a Lipschitz continuous Hessian assumption. He et al. (2025) analyzed momentum variants of normalized SGD and derived accelerated rates under higher order smoothness assumptions.

## 3. Problem Setup

Let $\mathcal{X} \subseteq \mathbb{R}^d$ and $\Xi$ denote the parameter space and the sample space, respectively. Consider a loss function $f$ :

$\mathcal{X} \times \Xi \to \mathbb{R}_+$. Let $S := \{\xi_1, \ldots, \xi_n\}$ be a dataset of $n$ i.i.d. samples drawn from a distribution $\mathcal{D}$ on $\Xi$. We are interested in finding a parameter $x \in \mathcal{X}$ that generalizes well, as measured by the population risk $F(x) := \mathbb{E}_{\xi \sim \mathcal{D}}[f(x; \xi)]$. Given a training algorithm $A$ on dataset $S$, it minimizes the empirical risk $F_S(x) := \frac{1}{n} \sum_{i=1}^n f(x; \xi_i)$, and we denote its output by $A(S)$. Let $\|\cdot\|$ denote the Euclidean norm, and $a \asymp b$ if $a$ and $b$ are equal up to constant factors. Let $\Delta := \mathbb{E}_{S,A}[F_S(x_0) - F_S^*]$, where $x_0$ is the initial point of algorithms and $F_S^* := \inf_x F_S(x)$. We define the clipping operator $\text{clip}_\gamma(\cdot)$ with parameter $\gamma$ as

$$\text{clip}_\gamma(u) = \begin{cases} u, & \|u\| \le \gamma, \\ \gamma \cdot \dfrac{u}{\|u\|}, & \|u\| > \gamma. \end{cases} \tag{3}$$

In the nonconvex setting, finding a global minimizer is NP-hard (Pardalos & Vavasis, 1991), and a learning algorithm can only guarantee to output an approximate stationary point, that is, we desire $\|\nabla F_S(A(S))\|$ to be small. In this case, the population risk is not an appropriate performance measure, and we instead use the population gradient norm $\|\nabla F(A(S))\|$, which can be decomposed as

$$\mathbb{E}_{S,A}[\|\nabla F(A(S))\|] \le \underbrace{\mathbb{E}_{S,A}[\|\nabla F_S(A(S))\|]}_{\text{optimization error}}$$
$$+ \underbrace{\mathbb{E}_{S,A}[\|\nabla F(A(S)) - \nabla F_S(A(S))\|]}_{\text{generalization error}}.$$

We will study the generalization error via algorithmic stability and derive optimization error bounds for specific algorithms under heavy-tailed noise.

We next present the assumptions used in our analysis. First, recall the $p$-BCM assumption on heavy-tailed noise, which is stated again as follows for convenience.

**Assumption 3.1.** There exist $p \in (1, 2]$ and $\sigma_p > 0$ such that for all $x$, it holds

$$\mathbb{E}_{\xi \sim \mathcal{D}}[\|\nabla f(x; \xi) - \nabla F(x)\|^p] \le \sigma_p^p.$$

Assumption 3.1 is standard in the literature on stochastic optimization under heavy-tailed noise (Zhang et al., 2020; Cutkosky & Mehta, 2021; Nguyen et al., 2023; Liu et al., 2023; Hübler et al., 2025). Another way to model heavy-tailed noise is to assume an $\alpha$-stable random variable $X$ with characteristic function $\mathbb{E}[\exp(i\omega X)] = \exp(-\sigma|\omega|^\alpha)$, where $i$ is the imaginary unit and $\omega \in \mathbb{R}$ (Simsekli et al., 2019). However, the $\alpha$-stable model can be restrictive, as it implicitly assumes that the noise is identically distributed across coordinates (Li et al., 2021b; Xie et al., 2021), while Assumption 3.1 imposes a weaker moment condition and is therefore more general. Additionally, recent work (Sun et al., 2025c) assumes an unbounded variance condition

of the form $\mathbb{E}_{\xi \sim \mathcal{D}}[\|\nabla f(x; \xi) - \nabla F(x)\|^2] \le c\|x\|^p + \sigma^2$, which does not directly model heavy-tailed noise. This condition and $p$-BCM cannot be derived from each other.

**Assumption 3.2.** For any $\xi$, the function $f(\cdot; \xi)$ is $L$-smooth, i.e., for any $x, y \in \mathbb{R}^d$, it holds

$$\|\nabla f(x; \xi) - \nabla f(y; \xi)\| \le L\|x - y\|.$$

The $L$-smoothness assumption is also standard in nonconvex stochastic optimization, and it will be used in our stability analysis of clipped SGD and normalized SGD.

# 4. Generalization Error Bound

In this section, we will establish a generalization error bound based on the algorithmic stability under heavy-tailed noise.

## 4.1. Algorithmic Stability

Algorithmic stability is a central concept in statistical learning theory, measuring the sensitivity of an algorithm to the perturbation of a single sample in the training dataset. We say $S$ and $S'$ are neighboring datasets if they differ by only a single sample.

**Definition 4.1.** A randomized algorithm $A$ is called $\epsilon$-*uniformly-argument-stable* if for all neighboring datasets $S$ and $S'$, one has

$$\mathbb{E}_A[\|A(S) - A(S')\|] \le \epsilon.$$

It is called $\epsilon$-*uniformly-stable in gradients* if for all neighboring datasets $S$ and $S'$, one has

$$\sup_{\xi \in \Xi} \mathbb{E}_A[\|\nabla f(A(S); \xi) - \nabla f(A(S'); \xi)\|^2] \le \epsilon^2.$$

Uniform argument stability (Bousquet & Elisseeff, 2002) directly measures the change in the algorithm output under a single sample perturbation, while uniform stability in gradients (Lei, 2023) measures stability through the discrepancy between gradients and is more suitable for nonconvex analyses based on stationary point. Under the $L$-smoothness assumption, it is straightforward to show that uniform argument stability implies uniform stability in gradients.

## 4.2. Generalization via Stability in Gradients

The core idea for establishing generalization via algorithmic stability is to introduce an independent ghost sample. Let $S' = \{\xi_1', \ldots, \xi_n'\}$ be drawn independently from $\mathcal{D}$. For each $i \in [n]$, define

$$S^{(i)} = \{\xi_1, \ldots, \xi_{i-1}, \xi_i', \xi_{i+1}, \ldots, \xi_n\}.$$

In the convex setting, the generalization error can be rewritten as an average of loss differences under the single replacement datasets $S^{(i)}$, that is,

$$\mathbb{E}_{S,A}\big[F_S(A(S)) - F(A(S))\big]$$
$$= \frac{1}{n}\sum_{i=1}^{n}\mathbb{E}_{S,A}\big[f(A(S);\xi_i) - f(A(S^{(i)});\xi_i)\big],$$

and is therefore directly bounded by the uniform stability in function values (Shalev-Shwartz et al., 2010; Hardt et al., 2016).

For nonconvex problems, a direct adaptation of the convex argument yields the population gradient bound

$$\mathbb{E}_{S,A}\big[\|\nabla F(A(S))\|\big] = \frac{1}{n}\sum_{i=1}^{n}\mathbb{E}\Big[\big\|\mathbb{E}_{\xi_i}\big[\nabla f(A(S^{(i)});\xi_i)\big]\big\|\Big],$$

whereas for the empirical risk one has

$$\|\nabla F_S(A(S))\| = \Big\|\frac{1}{n}\sum_{i=1}^{n}\nabla f(A(S);\xi_i)\Big\|.$$

Since the average over $i$ is outside the norm in the first equation but inside the norm in the second one, the convex argument cannot be directly applied to upper bound the nonconvex generalization error. Using the error decomposition of Bousquet et al. (2020), Lei (2023) showed that if the algorithm $A$ is $\epsilon$-uniformly stable in gradients, then

$$\mathbb{E}_{S,A}\big[\|\nabla F(A(S)) - \nabla F_S(A(S))\|\big] \le 4\epsilon + \sigma_2 n^{-\frac{1}{2}},$$

where $\sigma_2^2$ denotes the variance of the stochastic gradients. This nonconvex framework has been adopted in a range of subsequent studies (Chen et al., 2024; Zhang et al., 2024; Sun et al., 2025c).

However, the nonconvex generalization analysis in (Lei, 2023) relies on the bounded variance for stochastic gradients and thus does not cover heavy-tailed noise, which provides a more plausible description in neural network training (Simsekli et al., 2019; Zhang et al., 2020). Using a truncation argument for the stochastic gradients, we establish the following nonconvex generalization error bound under heavy-tailed noise in Theorem 4.2.

**Theorem 4.2.** *Let $A$ be $\epsilon$-uniformly stable in gradients and Assumption 3.1 holds. Then*

$$\mathbb{E}_{S,A}\big[\|\nabla F(A(S)) - \nabla F_S(A(S))\|\big] \le 4\epsilon + C_p\sigma_p n^{-\frac{p-1}{p}},$$

*where $C_p$ is a constant depending only on $p$, defined by $C_2 = 1$ and $C_p = \frac{p}{2(p-1)}\left(\frac{4(p-1)}{2-p}\right)^{\frac{2-p}{p}}$ for $p \in (1,2)$.*

*Remark* 4.3. Theorem 4.2 bounds the nonconvex generalization error in terms of uniform stability in gradients

under the $p$-BCM condition, extending the bounded variance result (Lei, 2023). When $p = 2$, the bound coincides with that in Lei (2023, Theorem 6). It is worth emphasizing that this extension is nontrivial, with the proof hinging on a truncation and tail decomposition. Moreover, for the constant $C_p$, it is straightforward to verify that $1 \le C_p \le 3$ and $\lim_{p\to 2^-} C_p = 1$.

Devroye et al. (2016) indicate that, for the empirical mean estimator, there exists a distribution with bounded $p$-th central moment such that the estimation error is at least of order $\Omega(n^{-(p-1)/p})$, which may suggest the term $n^{-(p-1)/p}$ is optimal. Deriving a lower bound for algorithmic stability-based generalization in nonconvex optimization is a challenging but interesting direction, and we leave it for future work.

### 4.3. Proof Sketch of Theorem 4.2

In this section, we outline the proof of Theorem 4.2. A complete proof is provided in Appendix B.

**Step 1: Decomposition and truncation.** Following the derivation in Lei (2023), the generalization error can be decomposed as

$$n\mathbb{E}_{S,A}\Big[\|\nabla F(A(S)) - \nabla F_S(A(S))\|\Big]$$
$$\le \sum_{i=1}^{n}\mathbb{E}_{S,A,\xi,\xi_i'}\Big[\|\nabla f(A(S);\xi) - \nabla f(A(S^{(i)});\xi)\|\Big]$$
$$+ \mathbb{E}_{S,A}\Big[\Big\|\sum_{i=1}^{n}\underbrace{\mathbb{E}_{\xi_i'}\big[\mathbb{E}_{\xi}[\nabla f(A(S^{(i)});\xi)] - \nabla f(A(S^{(i)});\xi_i)\big]}_{:=z_i(S)}\Big\|\Big]$$
$$+ \sum_{i=1}^{n}\mathbb{E}_{S,A,\xi_i'}\Big[\|\nabla f(A(S^{(i)});\xi_i) - \nabla f(A(S);\xi_i)\|\Big].$$

The first and the third terms can be bounded by $\epsilon$-uniform stability in gradients. Consequently, it holds that

$$n\mathbb{E}_{S,A}\Big[\|\nabla F(A(S)) - \nabla F_S(A(S))\|\Big]$$
$$\le 2n\epsilon + \mathbb{E}_{S,A}\Big[\Big\|\sum_{i=1}^{n}z_i(S)\Big\|\Big]. \tag{4}$$

The main technical challenge is to bound $\mathbb{E}\big[\|\sum_{i=1}^{n}z_i(S)\|\big]$. Under heavy-tailed noise, since $z_i(S)$ does not admit a bounded variance, one cannot proceed as in the existing analyses (Lei, 2023) by directly applying Jensen's inequality and expanding the squared norm of the sum.

To this end, we introduce the truncated stochastic gradient noise

$$T_\tau(x;\xi) := \mathrm{clip}_\tau\big(\nabla f(x;\xi) - \nabla F(x)\big),$$

where $\tau > 0$ is a parameter and we will later specify its optimal choice. Noting that $T_\tau(x;\xi)$ is not mean-zero, we

further define the centered truncated variable

$$\tilde{T}_\tau(x;\xi) := T_\tau(x;\xi) - M_\tau(x),$$

where $M_\tau(x) := \mathbb{E}_\xi\big[T_\tau(x;\xi)\big]$. We then decompose $z_i(S)$ into a truncated part and a residual tail:

$$z_i^{(\tau)}(S) := -\mathbb{E}_{\xi_i'}\Big[\tilde{T}_\tau\big(A(S^{(i)});\xi_i\big)\Big],$$

$$r_i^{(\tau)}(S) := z_i(S) - z_i^{(\tau)}(S).$$

As a result, we have

$$\Big\|\sum_{i=1}^n z_i(S)\Big\| \le \Big\|\sum_{i=1}^n z_i^{(\tau)}(S)\Big\| + \Big\|\sum_{i=1}^n r_i^{(\tau)}(S)\Big\|.$$

**Step 2: Bound $\mathbb{E}_{S,A}\Big[\big\|\sum_{i=1}^n z_i^{(\tau)}(S)\big\|\Big]$.** By Jensen's inequality, it holds that

$$\mathbb{E}_{S,A}\Big[\Big\|\sum_{i=1}^n z_i^{(\tau)}(S)\Big\|\Big] \le \sqrt{\mathbb{E}_{S,A}\Big[\Big\|\sum_{i=1}^n z_i^{(\tau)}(S)\Big\|^2\Big]}$$

$$\le \sqrt{\mathbb{E}\Big[\sum_{i=1}^n \|z_i^{(\tau)}(S)\|^2\Big]} + \sqrt{\mathbb{E}\Big[\sum_{i\ne j}\langle z_i^{(\tau)}(S), z_j^{(\tau)}(S)\rangle\Big]}.$$

For the diagonal terms, using $\|\mathrm{clip}_\tau(u)\|^2 \le \tau^{2-p}\|u\|^p$ for all $u$ and $p \in (1,2]$, together with Assumption 3.1, we obtain

$$\mathbb{E}_{S,A}\big[\|z_i^{(\tau)}(S)\|^2\big] \le \tau^{2-p}\sigma_p^p.$$

For the cross terms with $i \ne j$, we use the fact that $z_i^{(\tau)}(S)$ is constructed from a centered truncation and thus has zero mean. By introducing an additional perturbation of the dataset and applying $\epsilon$-uniform stability in gradients, one can derive

$$\mathbb{E}_{S,A}\big[\langle z_i^{(\tau)}(S), z_j^{(\tau)}(S)\rangle\big] \le 4\epsilon^2.$$

Combining the two bounds yields

$$\mathbb{E}_{S,A}\Big[\Big\|\sum_{i=1}^n z_i^{(\tau)}(S)\Big\|\Big] \le \sqrt{n\tau^{2-p}\sigma_p^p} + 2n\epsilon.$$

**Step 3: Bound $\mathbb{E}_{S,A}\Big[\big\|\sum_{i=1}^n r_i^{(\tau)}(S)\big\|\Big]$.** Recall that $r_i^{(\tau)}(S) = z_i(S) - z_i^{(\tau)}(S)$. By the definition of $z_i^{(\tau)}(S)$ via the truncated noise $\tilde{T}_\tau$, the residual $r_i^{(\tau)}(S)$ involves terms of the form $u - \mathrm{clip}_\tau(u)$ for some vector $u$. For any $u$ and $p \in (1,2]$, one has

$$\|u - \mathrm{clip}_\tau(u)\| \le \|u\|\mathbf{1}\{\|u\| > \tau\} \le \frac{\|u\|^p}{\tau^{p-1}}.$$

Combining this inequality with Assumption 3.1 yields

$$\mathbb{E}_{S,A}\big[\|r_i^{(\tau)}(S)\|\big] \le \frac{2\sigma_p^p}{\tau^{p-1}}.$$

**Step 4: Choose the optimal $\tau$.** Combining Steps 2 and 3 yields, for any $\tau > 0$, one has

$$\frac{1}{n}\mathbb{E}_{S,A}\Big[\Big\|\sum_{i=1}^n z_i(S)\Big\|\Big] \le \sqrt{\frac{\tau^{2-p}\sigma_p^p}{n}} + 2\epsilon + \frac{2\sigma_p^p}{\tau^{p-1}}. \quad (5)$$

When $p = 2$, letting $\tau \to +\infty$ gives

$$\frac{1}{n}\mathbb{E}_{S,A}\Big[\Big\|\sum_{i=1}^n z_i(S)\Big\|\Big] \le \frac{\sigma_2}{\sqrt{n}} + 2\epsilon.$$

When $p \in (1,2)$, optimizing over $\tau > 0$ in equation (5) yields the choice

$$\tau_* = \left(\frac{4(p-1)}{2-p}\sqrt{n\sigma_p^p}\right)^{\frac{2}{p}},$$

which gives

$$\frac{1}{n}\mathbb{E}_{S,A}\Big[\Big\|\sum_{i=1}^n z_i(S)\Big\|\Big] \le C_p\sigma_p n^{-\frac{p-1}{p}} + 2\epsilon.$$

Substituting the above bounds into equation (4) completes the proof of Theorem 4.2.

## 5. Applications to Algorithms

In this section, we apply the results in Section 4 and establish risk bounds for algorithms under heavy-tailed noise, focusing on two representative techniques, clipping and normalization. Specifically, we derive bounds for clipped SGD, mini-batch normalized SGD, normalized SGD with momentum, and normalized SGD with clipping and momentum.

### 5.1. Clipped SGD

Under heavy-tailed gradient noise, stochastic gradients may have large norms, leading to unstable updates and slow convergence. A common approach is gradient clipping, which replaces a large-norm stochastic gradient by a scaled version. Recall the clipping operator defined in equation (3). Clipped SGD (Algorithm 1) uses the original stochastic gradient when its norm is at most $\gamma$, and rescales it otherwise so that the norm equals $\gamma$. This controls the effect of extreme gradients, but introduces bias in the gradient estimate.

The $\epsilon$-uniform stability in gradients of clipped SGD is established as follows.

---

**Algorithm 1** Clipped SGD

---

**Require:** Initial point $x_0$, stepsizes $\{\eta_t\}_{t=0}^{T-1}$, clipping parameter $\gamma$
1: **for** $t = 0, 1, \ldots, T-1$ **do**
2:     Sample $i_t$ uniformly from $[n]$
3:     $x_{t+1} = x_t - \eta_t \mathrm{clip}_\gamma\big(\nabla f(x_t; \xi_{i_t})\big)$
4: **end for**
5: **Output:** Uniformly sample from $\{x_0, x_1, \ldots, x_{T-1}\}$

---

**Proposition 5.1** (Stability of clipped SGD). *Assume $x_0 = 0$ and Assumption 3.2 holds. Then Algorithm 1 is $\epsilon_c$-uniformly stable in gradients with*

$$\epsilon_c \leq 2L\gamma\sqrt{\frac{T}{n}}\sum_{t=0}^{T-1}\eta_t.$$

*Remark* 5.2. The assumption of $x_0 = 0$ is only for the ease of presentation, which is common in the analysis of stability (Lei, 2023; Schliserman & Koren, 2022). Note that our main results still hold without this assumption.

**Theorem 5.3** (Population gradient bound of clipped SGD). *Consider Algorithm 1 with a constant stepsize $\eta_t = \eta$ and clipping threshold $\gamma$. Assume $x_0 = 0$, Assumptions 3.1 and 3.2 hold, and there exists $G > 0$ such that $\mathbb{E}_{i_t}\big[\|\nabla f(x_t; \xi_{i_t})\|^p\big] \leq G^p$ for all $t$. Let $A(S)$ denote the output of Algorithm 1 on dataset $S$. Then we have*

$$\mathbb{E}_{S,A}\big[\|\nabla F(A(S))\|\big] \leq \sqrt{\frac{2\Delta}{\eta T} + G^p\gamma^{1-p} + \sqrt{L\eta G^p\gamma^{2-p}}}$$
$$+ 8L\gamma\eta T\sqrt{\frac{T}{n}} + C_p\sigma_p n^{-\frac{p-1}{p}},$$

*where $C_p$ is the constant in Theorem 4.2.*

**Corollary 5.4.** *Under the assumptions of Theorem 5.3, choose $T \asymp n^{\frac{1}{3}}$, $\eta \asymp n^{-\frac{p}{3(3p-2)}}$, and $\gamma \asymp n^{\frac{1}{3(3p-2)}}$. Then we have*

$$\mathbb{E}_{S,A}\big[\|\nabla F(A(S))\|\big] = O\Big(n^{-\frac{p-1}{3(3p-2)}}\Big).$$

### 5.2. Normalized SGD

Another widely used approach for handling heavy-tailed noise is gradient normalization. In contrast to clipping, which modifies only stochastic gradients with norms exceeding the threshold, normalized SGD normalizes the stochastic gradient at every iteration. It avoids tuning the clipping threshold and can attain optimal convergence rates, which has attracted substantial attention in recent works (Liu & Zhou, 2025; Hübler et al., 2025; Sun et al., 2025b).

We consider the mini-batch normalized SGD in Algorithm 2. When $B = 1$, Algorithm 2 reduces to vanilla normalized SGD.

---

**Algorithm 2** Mini-batch normalized SGD (NSGD-B)

---

**Require:** Initial point $x_0$, stepsizes $\{\eta_t\}_{t=0}^{T-1}$, batch size $B$
1: **for** $t = 0, 1, \ldots, T-1$ **do**
2:     Sample $i_t^{(1)}, \ldots, i_t^{(B)}$ i.i.d. uniformly from $[n]$
3:     $g_t = \frac{1}{B}\sum_{b=1}^{B}\nabla f(x_t; \xi_{i_t^{(b)}})$
4:     $x_{t+1} = x_t - \eta_t \cdot \dfrac{g_t}{\|g_t\|}$
5: **end for**
6: **Output:** Uniformly sample from $\{x_0, x_1, \ldots, x_{T-1}\}$

---

Since normalization discards gradient magnitude, iterates of vanilla normalized SGD can oscillate near an optimum and may even fail to converge, as observed in prior work (Hazan et al., 2015; Hübler et al., 2025). The following example illustrates this behavior. Consider the one-dimensional finite-sum problem with $n = 2$ and components

$$f_1(x) = \log(1 + e^x) \quad \text{and} \quad f_2(x) = \log(1 + e^{-x}).$$

A direct calculation shows that $F_S''(x) = \frac{e^x}{(1+e^x)^2} \in (0, 1/4]$ for all $x$, and $F_S'(0) = 0$, hence $x^* = 0$ is the unique global minimizer of $F_S$. Moreover, one has

$$\frac{f_1'(x)}{|f_1'(x)|} = 1 \quad \text{and} \quad \frac{f_2'(x)}{|f_2'(x)|} = -1.$$

With $i_t$ sampled uniformly from $\{1, 2\}$, the normalized update reduces to

$$x_t = x_0 - \sum_{k=0}^{t-1}\eta_k\zeta_k,$$

where $\{\zeta_t\}$ are i.i.d. Rademacher random variables with $\mathbb{P}(\zeta_t = 1) = \mathbb{P}(\zeta_t = -1) = 1/2$. If $\eta_t \equiv \eta > 0$, then $\{x_t\}$ is a simple random walk and does not converge to $x^*$.

This motivates choosing $B > 1$ in Algorithm 2, where averaging over a mini-batch stabilizes the normalized direction. We next bound its stability and derive the corresponding population gradient bound.

**Proposition 5.5** (Stability of NSGD-B). *Assume $x_0 = 0$ and Assumption 3.2 holds. Then Algorithm 2 is $\epsilon_b$-uniformly stable in gradients with*

$$\epsilon_b \leq 2L\sqrt{\frac{BT}{n}}\sum_{t=0}^{T-1}\eta_t.$$

*Remark* 5.6. For normalized SGD, heavy-tailed noise does not affect the stability bound, which is a key advantage of normalization. In contrast, the vanilla SGD $x_{t+1} = x_t - \eta g_t$ leads to $\|x_{t+1} - x_t\| = \eta_t\|g_t\|$, where $\|g_t\|$ can be extremely large under heavy-tailed noise that heavily affects the algorithmic stability.

**Theorem 5.7** (Population gradient bound of NSGD-B). *Consider Algorithm 2 with a constant stepsize $\eta_t = \eta$ and mini-batch size $B$. Assume $x_0 = 0$, Assumptions 3.1 and 3.2 hold, and there exists $G > 0$ such that $\mathbb{E}_{i_t^{(b)}}\big[\|\nabla f(x_t; \xi_{i_t^{(b)}})\|^p\big] \leq G^p$ for all $t$ and $b$. Let $A(S)$ denote the output of Algorithm 2 on dataset $S$. Then we have*

$$\mathbb{E}_{S,A}\big[\|\nabla F(A(S))\|\big] \leq \frac{\Delta}{\eta T} + \frac{L\eta}{2} + 8GB^{-\frac{p-1}{p}}$$
$$+ 8L\eta T\sqrt{\frac{BT}{n}} + C_p\sigma_p n^{-\frac{p-1}{p}},$$

*where $C_p$ is the constant in Theorem 4.2.*

*Remark* 5.8. The analysis for NSGD-B is based on Lemma A.3. This idea also works for other SGD variants in mini-batch setting. See Appendix D for details.

**Corollary 5.9.** *Under the assumptions of Theorem 5.7, choose $T \asymp n^{\frac{2(p-1)}{7p-6}}$, $\eta \asymp n^{-\frac{p-1}{7p-6}}$, and $B \asymp n^{\frac{p}{7p-6}}$. Then we have*

$$\mathbb{E}_{S,A}\big[\|\nabla F(A(S))\|\big] = O\Big(n^{-\frac{p-1}{7p-6}}\Big).$$

*Remark* 5.10. Compared with clipped SGD (Corollary 5.4), NSGD-B (Corollary 5.9) achieves a sharper population risk bound. The stability bound for clipped SGD scales linearly with $\gamma$, whereas the stability bound for NSGD-B scales with $\sqrt{B}$, which leads to a better result when balancing stability and optimization error.

*Remark* 5.11. The parameter choices in Corollary 5.9 are consistent with those obtained from the optimization perspective. In particular, these choices satisfy $\eta \asymp T^{-1/2}$ and $B \asymp T^{p/(2p-2)}$, which are the same as in (Hübler et al., 2025) for NSGD-B.

## 5.3. Normalized SGD with Momentum

---
**Algorithm 3** Normalized SGD with momentum (NSGD-M)

---
**Require:** Initial point $x_0$, stepsizes $\{\eta_t\}_{t=0}^{T-1}$, momentum parameter $\beta \in [0, 1)$, and $m_{-1} = 0$
1: **for** $t = 0, 1, \ldots, T-1$ **do**
2:     Sample $i_t$ uniformly from $[n]$
3:     $m_t = \beta m_{t-1} + (1-\beta)\nabla f(x_t; \xi_{i_t})$
4:     $x_{t+1} = x_t - \eta_t \cdot \dfrac{m_t}{\|m_t\|}$
5: **end for**
6: **Output:** Uniformly sample from $\{x_0, x_1, \ldots, x_{T-1}\}$

---

Another approach to address the possible nonconvergence of vanilla NSGD is to incorporate momentum. In contrast to NSGD-B, which increases the per-iteration sample size through independent sampling, NSGD-M (Algorithm 3) uses past stochastic gradients to make the gradient estimate more stable.

**Proposition 5.12** (Stability of NSGD-M). *Assume $x_0 = 0$ and Assumption 3.2 holds. Then Algorithm 3 is $\epsilon_m$-uniformly stable in gradients with*

$$\epsilon_m \leq 2L\sqrt{\frac{T}{n}}\sum_{t=0}^{T-1}\eta_t.$$

**Theorem 5.13** (Population gradient bound of NSGD-M). *Consider Algorithm 3 with a constant stepsize $\eta_t = \eta$. Assume $x_0 = 0$, Assumptions 3.1 and 3.2 hold, and there exists $G > 0$ such that $\mathbb{E}_{i_t}\big[\|\nabla f(x_t; \xi_{i_t})\|^p\big] \leq G^p$ for all $t$. Let $A(S)$ denote the output of Algorithm 3 on dataset $S$. Then we have*

$$\mathbb{E}_{S,A}\big[\|\nabla F(A(S))\|\big] \leq \frac{\Delta}{\eta T} + \frac{L\eta}{2} + 8G(1-\beta)^{\frac{p-1}{p}}$$
$$+ \frac{4G}{T} + \frac{4G\beta}{(1-\beta)T} + \frac{2L\eta\beta}{1-\beta} + 8L\eta T\sqrt{\frac{T}{n}} + C_p\sigma_p n^{-\frac{p-1}{p}},$$

*where $C_p$ is the constant in Theorem 4.2.*

**Corollary 5.14.** *Under the assumptions of Theorem 5.13, choose $T \asymp n^{\frac{3p-2}{7p-6}}$, $\eta \asymp n^{-\frac{2p-1}{7p-6}}$, and $1-\beta \asymp n^{-\frac{p}{7p-6}}$. Then we have*

$$\mathbb{E}_{S,A}\big[\|\nabla F(A(S))\|\big] = O\Big(n^{-\frac{p-1}{7p-6}}\Big).$$

*Remark* 5.15. NSGD-M (Corollary 5.14) and NSGD-B (Corollary 5.9) achieve the same risk bound in terms of $n$. Although the stability bound for NSGD-M does not involve the $\sqrt{B}$ factor, the momentum parameter $\beta$ limits further improvement of the rate. When $p = 2$, the rate in Corollary 5.14 reduces to $O(n^{-1/8})$, matching the result in (Sun et al., 2025c) for NSGD-M under the bounded variance assumption.

## 5.4. Normalized SGD with Clipping and Momentum

---
**Algorithm 4** Normalized SGD with clipping and momentum (NSGD-CM)

---
**Require:** Initial point $x_0$, stepsizes $\{\eta_t\}_{t=0}^{T-1}$, momentum parameter $\beta \in [0, 1)$, clipping parameter $\gamma$, $m_{-1} = 0$
1: **for** $t = 0, 1, \ldots, T-1$ **do**
2:     Sample $i_t$ uniformly from $[n]$
3:     $g_t = \text{clip}_\gamma\big(\nabla f(x_t; \xi_{i_t})\big)$
4:     $m_t = \beta m_{t-1} + (1-\beta)g_t$
5:     $x_{t+1} = x_t - \eta_t \cdot \dfrac{m_t}{\|m_t\|}$
6: **end for**
7: **Output:** Uniformly sample from $\{x_0, x_1, \ldots, x_{T-1}\}$

---

Cutkosky & Mehta (2021) introduced normalized SGD with clipping and momentum (Algorithm 4), which integrates clipping with momentum and gradient normalization, and this method was also studied in (Liu et al., 2023; Sun et al., 2025b).

**Proposition 5.16** (Stability of NSGD-CM). *Assume $x_0 = 0$ and Assumption 3.2 holds. Then Algorithm 4 is $\epsilon_{cm}$-uniformly stable in gradients with*

$$\epsilon_{cm} \leq 2L\sqrt{\frac{T}{n}} \sum_{t=0}^{T-1} \eta_t.$$

**Theorem 5.17** (Population gradient bound of NSGD-CM). *Consider Algorithm 4 with a constant stepsize $\eta_t = \eta$. Assume $x_0 = 0$, Assumptions 3.1 and 3.2 hold, and there exists $G > 0$ such that $\mathbb{E}_{i_t}\big[\|\nabla f(x_t; \xi_{i_t})\|^p\big] \leq G^p$ for all t. Let $A(S)$ denote the output of Algorithm 4 on dataset S. Then we have*

$$\mathbb{E}_{S,A}\big[\|\nabla F(A(S))\|\big] \leq \frac{\Delta}{\eta T} + \frac{L\eta}{2} + \frac{2L\eta}{1-\beta} + 2G^p\gamma^{1-p}$$
$$+ 8G(1-\beta)^{\frac{p-1}{p}} + \frac{4G}{T} + \frac{4G\beta}{(1-\beta)T}$$
$$+ 8L\eta T\sqrt{\frac{T}{n}} + C_p\sigma_p n^{-\frac{p-1}{p}},$$

*where $C_p$ is the constant in Theorem 4.2.*

**Corollary 5.18.** *Under the assumptions of Theorem 5.17, choose $T \asymp n^{\frac{3p-2}{7p-6}}$, $\eta \asymp n^{-\frac{2p-1}{7p-6}}$, $1 - \beta \asymp n^{-\frac{p}{7p-6}}$, and $\gamma \asymp n^{\frac{1}{7p-6}}$. Then we have*

$$\mathbb{E}_{S,A}\big[\|\nabla F(A(S))\|\big] = O\left(n^{-\frac{p-1}{7p-6}}\right).$$

*Remark* 5.19. Compared with clipped SGD (Corollary 5.4), incorporating normalization and momentum yields sharper stability and generalization bounds. In particular, the stability term no longer scales linearly with the clipping threshold $\gamma$. By contrast, introducing clipping into NSGD-M does not improve the population risk bound.

## 6. Conclusion

This paper studies stability based generalization for stochastic nonconvex optimization under heavy-tailed noise. Under the bounded $p$th centered moment condition, a generalization framework is established that links nonconvex generalization error to uniform stability in gradients through a truncation and tail decomposition, and it recovers the bounded variance result when $p = 2$. The framework is applied to representative methods for heavy-tailed optimization. For gradient clipping, the analysis characterizes how stability and population guarantees depend on the clipping threshold. For gradient normalization, we establish stability and risk bounds for the mini-batch and momentum variants of normalized SGD. For future work, it is interesting to study error bounds in the nonsmooth setting, derive high-probability generalization bounds, and develop lower bounds for learning under heavy-tailed noise.

## Acknowledgments

This work was supported in part by the National Natural Science Foundation of China (No. 12571557), the Major Key Project of Pengcheng Laboratory (No. PCL2024A06), the Shanghai Basic Research Program (23JC1401000), and the National Key R&D Program of China (No. 2023YFA1009300). The work of X. Yuan was supported by the Hong Kong Research Grants Council under the GRF project 17309824 and Croucher Senior Fellowship.

## Impact Statement

This paper presents work whose goal is to advance the field of machine learning. There are many potential societal consequences of our work, none of which we feel must be specifically highlighted here.

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

## A. Useful Lemmas

In this section, we present several lemmas that will be repeatedly used in the subsequent proofs. We begin with basic properties of the clipping operator.

**Lemma A.1** (Basic properties of the clipping operator). *Let $\gamma > 0$ and define $\mathrm{clip}_\gamma(\cdot)$ by equation (3). For any $p \in (1, 2]$, the following properties hold.*

*(a) For any $u \in \mathbb{R}^d$, one has*

$$\|\mathrm{clip}_\gamma(u)\|^2 \leq \gamma^{2-p}\|u\|^p.$$

*(b) For any $u \in \mathbb{R}^d$, one has*

$$\|u - \mathrm{clip}_\gamma(u)\| \leq \frac{\|u\|^p}{\gamma^{p-1}}.$$

*(c) The mapping $\mathrm{clip}_\gamma(\cdot)$ is 1-Lipschitz, i.e., for any $u, v \in \mathbb{R}^d$, it holds*

$$\|\mathrm{clip}_\gamma(u) - \mathrm{clip}_\gamma(v)\| \leq \|u - v\|.$$

*Proof.* For statement (a), note that $\|\mathrm{clip}_\gamma(u)\| = \min\{\|u\|, \gamma\}$. If $\|u\| \leq \gamma$, then it holds

$$\|\mathrm{clip}_\gamma(u)\|^2 = \|u\|^2 = \|u\|^p\|u\|^{2-p} \leq \gamma^{2-p}\|u\|^p.$$

If $\|u\| > \gamma$, then

$$\|\mathrm{clip}_\gamma(u)\|^2 = \gamma^2 = \gamma^{2-p}\gamma^p \leq \gamma^{2-p}\|u\|^p.$$

For statement (b), the claim is trivial when $\|u\| \leq \gamma$. If $\|u\| > \gamma$, then $\mathrm{clip}_\gamma(u) = \gamma u/\|u\|$. Hence, it holds

$$\|u - \mathrm{clip}_\gamma(u)\| = \left\|u - \gamma\frac{u}{\|u\|}\right\| = \|u\| - \gamma \leq \|u\| = \frac{\|u\|^p}{\gamma^{p-1}}\left(\frac{\gamma}{\|u\|}\right)^{p-1} \leq \frac{\|u\|^p}{\gamma^{p-1}},$$

where the last inequality uses $\|u\| > \gamma$ and $p > 1$.

For statement (c), we prove $\|\mathrm{clip}_\gamma(u) - \mathrm{clip}_\gamma(v)\| \leq \|u - v\|$ by a case analysis.

If $\|u\| \leq \gamma$ and $\|v\| \leq \gamma$, then $\mathrm{clip}_\gamma(u) = u$ and $\mathrm{clip}_\gamma(v) = v$, and the claim follows.

If $\|u\| \leq \gamma < \|v\|$, then $\mathrm{clip}_\gamma(u) = u$ and $\mathrm{clip}_\gamma(v) = \gamma v/\|v\|$. Expanding and simplifying, we have

$$\|u - v\|^2 - \left\|u - \gamma\frac{v}{\|v\|}\right\|^2 = \|v\|^2 - \gamma^2 - 2\left(1 - \frac{\gamma}{\|v\|}\right)\langle u, v\rangle$$

$$= (\|v\| - \gamma)\left(\|v\| + \gamma - 2\frac{\langle u, v\rangle}{\|v\|}\right).$$

Since $\langle u, v\rangle/\|v\| \leq \|u\| \leq \gamma$, the second factor is at least $\|v\| - \gamma > 0$, and hence the right-hand side is nonnegative. This implies $\left\|u - \gamma\frac{v}{\|v\|}\right\| \leq \|u - v\|$. The case $\|v\| \leq \gamma < \|u\|$ follows by symmetry.

If $\|u\| > \gamma$ and $\|v\| > \gamma$, then $\mathrm{clip}_\gamma(u) = \gamma u/\|u\|$ and $\mathrm{clip}_\gamma(v) = \gamma v/\|v\|$. Expanding both sides gives

$$\|u - v\|^2 - \left\|\gamma\frac{u}{\|u\|} - \gamma\frac{v}{\|v\|}\right\|^2 = \|u\|^2 + \|v\|^2 - 2\gamma^2 - 2\langle u, v\rangle\left(1 - \frac{\gamma^2}{\|u\|\|v\|}\right).$$

Here $\|u\|\|v\| > \gamma^2$, so $1 - \gamma^2/(\|u\|\|v\|) > 0$. Using $\langle u, v\rangle \leq \|u\|\|v\|$, we obtain

$$-2\langle u, v\rangle\left(1 - \frac{\gamma^2}{\|u\|\|v\|}\right) \geq -2\|u\|\|v\| + 2\gamma^2,$$

and hence the difference is at least $\|u\|^2 + \|v\|^2 - 2\|u\|\|v\| = (\|u\| - \|v\|)^2 \geq 0$. This implies $\mathrm{clip}_\gamma(\cdot)$ is 1-Lipschitz. $\square$

The following Lemmas A.2 and A.3 play an important role in the analysis of normalized methods.

**Lemma A.2.** *(Hübler et al., 2025, Lemma 7) For any $u, v \in \mathbb{R}^d$ with $v \neq 0$, it holds that*

$$\left\langle u, \frac{v}{\|v\|} \right\rangle \geq \|u\| - 2\|u - v\|.$$

**Lemma A.3.** *(Hübler et al., 2025, Lemma 10) Let $p \in (1, 2]$, and let $X_1, \ldots, X_n \in \mathbb{R}^d$ be a martingale difference sequence. Then we have*

$$\mathbb{E}\left[ \left\| \sum_{i=1}^{n} X_i \right\|^p \right] \leq 2 \sum_{i=1}^{n} \mathbb{E}\left[ \|X_i\|^p \right].$$

## B. The Proof of Theorem 4.2

Let $S' = \{\xi_1', \ldots, \xi_n'\}$ be drawn independently from $\mathcal{D}$. For each $i \in \{1, \ldots, n\}$, define

$$S^{(i)} = \{\xi_1, \ldots, \xi_{i-1}, \xi_i', \xi_{i+1}, \ldots, \xi_n\}.$$

**Step 1: Decomposition and truncation.** Following existing works (Bousquet et al., 2020; Lei, 2023), we have the following decomposition

$$
\begin{aligned}
n\Big(\nabla F(A(S)) - \nabla F_S(A(S))\Big) &= n\mathbb{E}_\xi\big[\nabla f(A(S); \xi)\big] - \sum_{i=1}^{n} \nabla f(A(S); \xi_i) \\
&= \sum_{i=1}^{n} \mathbb{E}_{\xi, \xi_i'}\Big[\nabla f\big(A(S); \xi\big) - \nabla f\big(A(S^{(i)}); \xi\big)\Big] \\
&\quad + \sum_{i=1}^{n} \mathbb{E}_{\xi_i'}\Big[\mathbb{E}_\xi\big[\nabla f\big(A(S^{(i)}); \xi\big)\big] - \nabla f\big(A(S^{(i)}); \xi_i\big)\Big] \\
&\quad + \sum_{i=1}^{n} \mathbb{E}_{\xi_i'}\Big[\nabla f\big(A(S^{(i)}); \xi_i\big) - \nabla f\big(A(S); \xi_i\big)\Big].
\end{aligned}
\tag{6}
$$

We define

$$z_i(S) := \mathbb{E}_{\xi_i'}\Big[\mathbb{E}_\xi\big[\nabla f\big(A(S^{(i)}); \xi\big)\big] - \nabla f\big(A(S^{(i)}); \xi_i\big)\Big].$$

Taking norms on both sides of equation (6), applying the triangle inequality, and then taking expectation yield

$$
\begin{aligned}
n\mathbb{E}_{S,A}\Big[\big\|\nabla F(A(S)) - \nabla F_S(A(S))\big\|\Big] &\leq \sum_{i=1}^{n} \mathbb{E}_{S,A,\xi,\xi_i'}\Big[\big\|\nabla f\big(A(S); \xi\big) - \nabla f\big(A(S^{(i)}); \xi\big)\big\|\Big] \\
&\quad + \mathbb{E}_{S,A}\Big[\Big\|\sum_{i=1}^{n} z_i(S)\Big\|\Big] \\
&\quad + \sum_{i=1}^{n} \mathbb{E}_{S,A,\xi_i'}\Big[\big\|\nabla f\big(A(S^{(i)}); \xi_i\big) - \nabla f\big(A(S); \xi_i\big)\big\|\Big].
\end{aligned}
$$

By Jensen's inequality and Cauchy–Schwarz, together with the $\epsilon$-uniform stability in gradients, each term in the first and third sums is bounded by $\epsilon$, which gives

$$n\mathbb{E}_{S,A}\Big[\big\|\nabla F(A(S)) - \nabla F_S(A(S))\big\|\Big] \leq 2n\epsilon + \mathbb{E}_{S,A}\Big[\Big\|\sum_{i=1}^{n} z_i(S)\Big\|\Big]. \tag{7}$$

Then we will bound $\mathbb{E}_{S,A}\Big[\big\|\sum_{i=1}^{n} z_i(S)\big\|\Big]$. For $\tau > 0$, we define

$$T_\tau(x; \xi) := \text{clip}_\tau\big(\nabla f(x; \xi) - \nabla F(x)\big), \qquad M_\tau(x) := \mathbb{E}_\xi\big[T_\tau(x; \xi)\big], \qquad \tilde{T}_\tau(x; \xi) := T_\tau(x; \xi) - M_\tau(x),$$

$$z_i^{(\tau)}(S) := -\mathbb{E}_{\xi_i'}\Big[\tilde{T}_\tau\big(A(S^{(i)}); \xi_i\big)\Big], \quad \text{and} \quad r_i^{(\tau)}(S) := z_i(S) - z_i^{(\tau)}(S).$$

Then we have

$$\Big\|\sum_{i=1}^n z_i(S)\Big\| \le \Big\|\sum_{i=1}^n z_i^{(\tau)}(S)\Big\| + \Big\|\sum_{i=1}^n r_i^{(\tau)}(S)\Big\|.$$

**Step 2: Bound** $\mathbb{E}_{S,A}\Big[\Big\|\sum_{i=1}^n z_i^{(\tau)}(S)\Big\|\Big].$

Note that $\mathbb{E}_{\xi_i}\big[z_i^{(\tau)}(S)\big] = 0$. By expanding the squared norm, we have

$$\mathbb{E}_{S,A}\Big[\Big\|\sum_{i=1}^n z_i^{(\tau)}(S)\Big\|^2\Big] = \mathbb{E}_{S,A}\Big[\sum_{i=1}^n \big\|z_i^{(\tau)}(S)\big\|^2\Big] + \mathbb{E}_{S,A}\Big[\sum_{i\ne j}\langle z_i^{(\tau)}(S), z_j^{(\tau)}(S)\rangle\Big].$$

Let $S'' = \{\xi_1'', \ldots, \xi_n''\}$, where $\xi_k'' \sim \mathcal{D}$ i.i.d. Define

$$S_j := \{\xi_1, \ldots, \xi_{j-1}, \xi_j'', \xi_{j+1}, \ldots, \xi_n\},$$
$$S_j^{(i)} := \{\xi_1, \ldots, \xi_{i-1}, \xi_i', \xi_{i+1}, \ldots, \xi_{j-1}, \xi_j'', \xi_{j+1}, \ldots, \xi_n\}.$$

We first bound $\mathbb{E}_{S,A}\big[\langle z_i^{(\tau)}(S), z_j^{(\tau)}(S)\rangle\big]$ for $i \ne j$. Using the fact that $\mathbb{E}_{\xi_j}\big[z_j^{(\tau)}(S)\big] = 0$, we have

$$\mathbb{E}_{S,A}\big[\langle z_i^{(\tau)}(S_j), z_j^{(\tau)}(S)\rangle\big] = \mathbb{E}_{S,A}\Big[\mathbb{E}_{\xi_j}\big[\langle z_i^{(\tau)}(S_j), z_j^{(\tau)}(S)\rangle\big]\Big] = 0.$$

Similarly, we have

$$\mathbb{E}_{S,A}\big[\langle z_i^{(\tau)}(S), z_j^{(\tau)}(S_i)\rangle\big] = 0 \quad \text{and} \quad \mathbb{E}_{S,A}\big[\langle z_i^{(\tau)}(S_j), z_j^{(\tau)}(S_i)\rangle\big] = 0.$$

Therefore,

$$\mathbb{E}_{S,A}\big[\langle z_i^{(\tau)}(S), z_j^{(\tau)}(S)\rangle\big] = \mathbb{E}_{S,A}\Big[\big\langle z_i^{(\tau)}(S) - z_i^{(\tau)}(S_j),\ z_j^{(\tau)}(S) - z_j^{(\tau)}(S_i)\big\rangle\Big]$$
$$\le \frac{1}{2}\mathbb{E}_{S,A}\big[\|z_i^{(\tau)}(S) - z_i^{(\tau)}(S_j)\|^2\big] + \frac{1}{2}\mathbb{E}_{S,A}\big[\|z_j^{(\tau)}(S) - z_j^{(\tau)}(S_i)\|^2\big],$$

where we used $\langle a, b\rangle \le \frac{1}{2}\|a\|^2 + \frac{1}{2}\|b\|^2$.

Moreover, by Jensen's inequality, we have

$$\|z_i^{(\tau)}(S) - z_i^{(\tau)}(S_j)\|^2 = \Big\|\mathbb{E}_{\xi_i'}\big[\tilde{T}_\tau(A(S^{(i)}); \xi_i) - \tilde{T}_\tau(A(S_j^{(i)}); \xi_i)\big]\Big\|^2$$
$$\le \mathbb{E}_{\xi_i'}\big[\|\tilde{T}_\tau(A(S^{(i)}); \xi_i) - \tilde{T}_\tau(A(S_j^{(i)}); \xi_i)\|^2\big].$$

Furthermore, one has

$$\mathbb{E}_{\xi_i}\big[\|\tilde{T}_\tau(A(S^{(i)}); \xi_i) - \tilde{T}_\tau(A(S_j^{(i)}); \xi_i)\|^2\big] \le 4\mathbb{E}_{\xi_i}\big[\|T_\tau(A(S^{(i)}); \xi_i) - T_\tau(A(S_j^{(i)}); \xi_i)\|^2\big],$$

where we used $\|u + v\|^2 \le 2\|u\|^2 + 2\|v\|^2$ and Jensen's inequality for $M_\tau(x) = \mathbb{E}_\xi[T_\tau(x; \xi)]$.

By the definition of $T_\tau$ and Lemma A.1 (c), we have

$$\|T_\tau(A(S^{(i)}); \xi_i) - T_\tau(A(S_j^{(i)}); \xi_i)\|^2$$
$$\le \big\|(\nabla f(A(S^{(i)}); \xi_i) - \nabla F(A(S^{(i)}))) - (\nabla f(A(S_j^{(i)}); \xi_i) - \nabla F(A(S_j^{(i)})))\big\|^2.$$

Taking expectation over $\xi_i$ and using $\mathbb{E}\|X - \mathbb{E}X\|^2 \le \mathbb{E}\|X\|^2$ yield

$$\mathbb{E}_{\xi_i}\big[\|T_\tau(A(S^{(i)}); \xi_i) - T_\tau(A(S_j^{(i)}); \xi_i)\|^2\big] \le \mathbb{E}_{\xi_i}\big[\|\nabla f(A(S^{(i)}); \xi_i) - \nabla f(A(S_j^{(i)}); \xi_i)\|^2\big].$$

Consequently, it holds

$$\mathbb{E}_{S,A}\big[\langle z_i^{(\tau)}(S), z_j^{(\tau)}(S)\rangle\big] \le 4\mathbb{E}_{S,A}\mathbb{E}_{\xi_i'}\mathbb{E}_{\xi_i}\big[\|\nabla f(A(S^{(i)}); \xi_i) - \nabla f(A(S_j^{(i)}); \xi_i)\|^2\big]$$
$$\le 4\epsilon^2,$$

where the last inequality follows from the $\epsilon$-uniform stability in gradients, since $S^{(i)}$ and $S_j^{(i)}$ are neighboring datasets.

Then we bound $\mathbb{E}_{S,A}\big[\sum_{i=1}^n \|z_i^{(\tau)}(S)\|^2\big]$. By Jensen's inequality, we have

$$\mathbb{E}_{S,A}\big[\|z_i^{(\tau)}(S)\|^2\big] \le \mathbb{E}_{S,A}\mathbb{E}_{\xi_i'}\big[\|\tilde{T}_\tau(A(S^{(i)}); \xi_i)\|^2\big]$$
$$= \mathbb{E}_{S,A}\mathbb{E}_{\xi_i'}\big[\|T_\tau(A(S^{(i)}); \xi_i) - M_\tau(A(S^{(i)}))\|^2\big]$$
$$\le \mathbb{E}_{S,A}\mathbb{E}_{\xi_i'}\big[\|T_\tau(A(S^{(i)}); \xi_i)\|^2\big],$$

where the last inequality uses $\mathbb{E}\|X - \mathbb{E}X\|^2 \le \mathbb{E}\|X\|^2$.

By Lemma A.1 (a), it holds that

$$\|T_\tau(A(S^{(i)}); \xi_i)\|^2 = \big\|\mathrm{clip}_\tau\big(\nabla f(A(S^{(i)}); \xi_i) - \nabla F(A(S^{(i)}))\big)\big\|^2$$
$$\le \tau^{2-p}\big\|\nabla f(A(S^{(i)}); \xi_i) - \nabla F(A(S^{(i)}))\big\|^p.$$

Consequently, it holds

$$\mathbb{E}_{S,A}\big[\|z_i^{(\tau)}(S)\|^2\big] \le \tau^{2-p}\mathbb{E}_{S,A}\mathbb{E}_{\xi_i'}\mathbb{E}_{\xi_i}\big[\|\nabla f(A(S^{(i)}); \xi_i) - \nabla F(A(S^{(i)}))\|^p\big]$$
$$\le \tau^{2-p}\sigma_p^p, \tag{8}$$

where the last inequality follows from Assumption 3.1.

Combining equation (8) and the previous estimate

$$\mathbb{E}_{S,A}\big[\langle z_i^{(\tau)}(S), z_j^{(\tau)}(S)\rangle\big] \le 4\epsilon^2, \qquad i \ne j,$$

we obtain

$$\mathbb{E}\Big[\big\|\sum_{i=1}^n z_i^{(\tau)}(S)\big\|\Big] \le \sqrt{\mathbb{E}\Big[\big\|\sum_{i=1}^n z_i^{(\tau)}(S)\big\|^2\Big]} \le \sqrt{n\tau^{2-p}\sigma_p^p + 4n(n-1)\epsilon^2},$$

where the first inequality uses Jensen's inequality.

Dividing both sides by $n$ yields

$$\frac{1}{n}\mathbb{E}\Big[\big\|\sum_{i=1}^n z_i^{(\tau)}(S)\big\|\Big] \le \sqrt{\frac{\tau^{2-p}\sigma_p^p}{n} + 4\frac{n-1}{n}\epsilon^2} \le \sqrt{\frac{\tau^{2-p}\sigma_p^p}{n}} + 2\epsilon, \tag{9}$$

where we used $\sqrt{a+b} \le \sqrt{a} + \sqrt{b}$ and $\sqrt{4\frac{n-1}{n}\epsilon^2} \le 2\epsilon$.

**Step 3: Bound $\mathbb{E}_{S,A}\Big[\big\|\sum_{i=1}^n r_i^{(\tau)}(S)\big\|\Big]$.**

Recall that

$$r_i^{(\tau)}(S) = z_i(S) - z_i^{(\tau)}(S)$$

$$= -\mathbb{E}_{\xi_i'}\Big[\big(\nabla f(A(S^{(i)});\xi_i) - \nabla F(A(S^{(i)}))\big) - \big(T_\tau(A(S^{(i)});\xi_i) - M_\tau(A(S^{(i)}))\big)\Big]$$

$$= -\mathbb{E}_{\xi_i'}\Big[\big(\nabla f(A(S^{(i)});\xi_i) - \nabla F(A(S^{(i)}))\big) - T_\tau(A(S^{(i)});\xi_i)\Big]$$

$$+ \mathbb{E}_{\xi_i'}\mathbb{E}_\xi\Big[T_\tau(A(S^{(i)});\xi) - \big(\nabla f(A(S^{(i)});\xi) - \nabla F(A(S^{(i)}))\big)\Big].$$

Therefore, by the triangle inequality and Jensen's inequality,

$$\mathbb{E}_{\xi_i}\big[\|r_i^{(\tau)}(S)\|\big] \leq 2\mathbb{E}_{\xi_i'}\mathbb{E}_\xi\Big[\big\|\big(\nabla f(A(S^{(i)});\xi) - \nabla F(A(S^{(i)}))\big) - T_\tau(A(S^{(i)});\xi)\big\|\Big].$$

Applying Lemma A.1 (b) with $u = \nabla f(A(S^{(i)});\xi) - \nabla F(A(S^{(i)}))$, we obtain

$$\mathbb{E}_{S,A}\big[\|r_i^{(\tau)}(S)\|\big] \leq \frac{2}{\tau^{p-1}}\mathbb{E}_{S,A}\mathbb{E}_{\xi_i'}\mathbb{E}_\xi\Big[\big\|\nabla f(A(S^{(i)});\xi) - \nabla F(A(S^{(i)}))\big\|^p\Big]$$

$$\leq \frac{2\sigma_p^p}{\tau^{p-1}}, \tag{10}$$

where the last inequality follows from Assumption 3.1.

**Step 4: Choose the optimal $\tau$.** Combining the bounds on $\big\|\sum_{i=1}^n z_i^{(\tau)}(S)\big\|$ and $\big\|\sum_{i=1}^n r_i^{(\tau)}(S)\big\|$ (equation (9) and equation (10)), by the triangle inequality and Jensen's inequality, we have

$$\frac{1}{n}\mathbb{E}_{S,A}\Big[\big\|\sum_{i=1}^n z_i(S)\big\|\Big] \leq \frac{1}{n}\mathbb{E}_{S,A}\Big[\big\|\sum_{i=1}^n z_i^{(\tau)}(S)\big\|\Big] + \frac{1}{n}\mathbb{E}_{S,A}\Big[\big\|\sum_{i=1}^n r_i^{(\tau)}(S)\big\|\Big]$$

$$\leq \sqrt{\frac{\tau^{2-p}\sigma_p^p}{n}} + 2\epsilon + \frac{2\sigma_p^p}{\tau^{p-1}}.$$

If $p = 2$, letting $\tau \to \infty$ yields

$$\frac{1}{n}\mathbb{E}_{S,A}\Big[\big\|\sum_{i=1}^n z_i(S)\big\|\Big] \leq \frac{\sigma_2}{\sqrt{n}} + 2\epsilon.$$

If $p \in (1,2)$, define

$$\phi(\tau) := \sqrt{\frac{\sigma_p^p}{n}\tau^{\frac{2-p}{2}}} + 2\sigma_p^p\tau^{1-p}.$$

Then

$$\phi'(\tau) = \sqrt{\frac{\sigma_p^p}{n}} \cdot \frac{2-p}{2}\tau^{-\frac{p}{2}} + 2\sigma_p^p(1-p)\tau^{-p}.$$

Setting $\phi'(\tau) = 0$ gives

$$\tau_* = \left(\frac{4(p-1)}{2-p}\sqrt{n\sigma_p^p}\right)^{\frac{2}{p}}.$$

Substituting $\tau_*$ yields

$$\phi(\tau_*) = \frac{p}{2(p-1)}\left(\frac{4(p-1)}{2-p}\right)^{\frac{2-p}{p}}\sigma_p n^{-\frac{p-1}{p}} = C_p\sigma_p n^{-\frac{p-1}{p}}.$$

Therefore, for $p \in (1,2]$, it holds

$$\frac{1}{n}\mathbb{E}_{S,A}\Big[\big\|\sum_{i=1}^n z_i(S)\big\|\Big] \leq C_p\sigma_p n^{-\frac{p-1}{p}} + 2\epsilon. \tag{11}$$

Substituting equation (11) into equation (7) completes the proof.

# C. The Proofs for Section 5

In this section, we provide the proofs of propositions and theorems in Section 5.

## C.1. The Proof of Proposition 5.1

Without loss of generality, assume $S$ and $S'$ differ only at the $n$-th sample, i.e., $\xi_n \neq \xi'_n$. By Assumption 3.2, for any $\xi$,

$$\mathbb{E}_A\left[\left\|\nabla f(A(S); \xi) - \nabla f(A(S'); \xi)\right\|^2\right] \leq L^2 \mathbb{E}_A\left[\|A(S) - A(S')\|^2\right]. \tag{12}$$

Let $\{x_t\}_{t=0}^T$ and $\{x'_t\}_{t=0}^T$ be the iterates of Algorithm 1 on $S$ and $S'$, respectively, with $x_0 = x'_0 = 0$. By the update rule and $\|\mathrm{clip}_\gamma(v)\| \leq \gamma$, we have for all $t$,

$$\|x_{t+1} - x_t\| = \eta_t \left\|\mathrm{clip}_\gamma(\nabla f(x_t; \xi_{i_t}))\right\| \leq \gamma \eta_t.$$

Hence, by the triangle inequality, we have

$$\|x_t\| \leq \gamma \sum_{s=0}^{t-1} \eta_s \quad \text{and} \quad \|x'_t\| \leq \gamma \sum_{s=0}^{t-1} \eta_s.$$

Therefore, it holds that

$$\|A(S)\| \leq \gamma \sum_{t=0}^{T-1} \eta_t \quad \text{and} \quad \|A(S')\| \leq \gamma \sum_{t=0}^{T-1} \eta_t.$$

Let $I(A) := \{i_t\}_{t=0}^{T-1}$ be the index sequence sampled by the algorithm. It holds that $A(S) = A(S')$ on the event $\{n \notin I(A)\}$, and hence

$$
\begin{aligned}
\mathbb{E}_A\left[\|A(S) - A(S')\|^2\right] &= \mathbb{E}_A\left[\|A(S) - A(S')\|^2 \mid n \notin I(A)\right]\mathbb{P}(n \notin I(A)) \\
&\quad + \mathbb{E}_A\left[\|A(S) - A(S')\|^2 \mid n \in I(A)\right]\mathbb{P}(n \in I(A)) \\
&= \mathbb{E}_A\left[\|A(S) - A(S')\|^2 \mid n \in I(A)\right]\mathbb{P}(n \in I(A)) \\
&\leq 2\gamma^2\left(\sum_{t=0}^{T-1}\eta_t\right)^2 \mathbb{P}(n \in I(A)) + 2\gamma^2\left(\sum_{t=0}^{T-1}\eta_t\right)^2 \mathbb{P}(n \in I(A)) \\
&\leq 4\gamma^2\left(\sum_{t=0}^{T-1}\eta_t\right)^2 \mathbb{P}(n \in I(A)).
\end{aligned}
$$

Since $i_t$ is sampled uniformly from $[n]$ at each iteration, we have

$$\mathbb{P}(n \in I(A)) \leq \sum_{t=0}^{T-1} \mathbb{P}(i_t = n) = \frac{T}{n}.$$

Combining the above bounds yields

$$\mathbb{E}_A\left[\|A(S) - A(S')\|^2\right] \leq 4\gamma^2\left(\sum_{t=0}^{T-1}\eta_t\right)^2 \frac{T}{n}. \tag{13}$$

Substituting equation (13) into equation (12) completes the proof.

## C.2. The Proof of Theorem 5.3

By Theorem 4.2 and Proposition 5.1, we have

$$\mathbb{E}_{S,A}\left[\|\nabla F(A(S))\|\right] \leq \mathbb{E}_{S,A}\left[\|\nabla F_S(A(S))\|\right] + 8L\gamma\eta T\sqrt{\frac{T}{n}} + C_p\sigma_p n^{-\frac{p-1}{p}}.$$

We next establish an upper bound on the optimization error of clipped SGD. Let $\mathcal{F}_t$ be the sigma-field generated by $\{S,\ x_0,\ i_0, \ldots, i_{t-1}\}$. Define $g_t := \mathrm{clip}_\gamma(\nabla f(x_t; \xi_{i_t}))$. By $L$-smoothness of $F_S$, we have

$$F_S(x_{t+1}) \le F_S(x_t) + \langle \nabla F_S(x_t), x_{t+1} - x_t \rangle + \frac{L}{2}\|x_{t+1} - x_t\|^2$$
$$= F_S(x_t) - \eta \langle \nabla F_S(x_t), g_t \rangle + \frac{L\eta^2}{2}\|g_t\|^2.$$

Taking conditional expectation given $\mathcal{F}_t$ yields

$$\mathbb{E}\big[F_S(x_{t+1}) \mid \mathcal{F}_t\big] \le F_S(x_t) - \eta \langle \nabla F_S(x_t), \mathbb{E}[g_t \mid \mathcal{F}_t] \rangle + \frac{L\eta^2}{2}\mathbb{E}\big[\|g_t\|^2 \mid \mathcal{F}_t\big]$$
$$= F_S(x_t) - \eta\|\nabla F_S(x_t)\|^2 - \eta\langle \nabla F_S(x_t), \mathbb{E}[g_t \mid \mathcal{F}_t] - \nabla F_S(x_t) \rangle + \frac{L\eta^2}{2}\mathbb{E}\big[\|g_t\|^2 \mid \mathcal{F}_t\big]$$
$$\le F_S(x_t) - \frac{\eta}{2}\|\nabla F_S(x_t)\|^2 + \frac{\eta}{2}\big\|\mathbb{E}[g_t \mid \mathcal{F}_t] - \nabla F_S(x_t)\big\|^2 + \frac{L\eta^2}{2}\mathbb{E}\big[\|g_t\|^2 \mid \mathcal{F}_t\big],$$

where the last inequality uses $-\langle a, b \rangle \le \frac{1}{2}\|a\|^2 + \frac{1}{2}\|b\|^2$.

Taking expectation and summing over $t = 0, \ldots, T-1$, we obtain

$$\sum_{t=0}^{T-1} \frac{\eta}{2}\mathbb{E}\big[\|\nabla F_S(x_t)\|^2\big] \le \mathbb{E}\big[F_S(x_0) - F_S(x_T)\big] + \sum_{t=0}^{T-1} \frac{\eta}{2}\mathbb{E}\big[\|\mathbb{E}[g_t \mid \mathcal{F}_t] - \nabla F_S(x_t)\|^2\big]$$
$$+ \sum_{t=0}^{T-1} \frac{L\eta^2}{2}\mathbb{E}\big[\|g_t\|^2\big]$$
$$\le \Delta + \sum_{t=0}^{T-1} \frac{\eta}{2}\mathbb{E}\big[\|\mathbb{E}[g_t \mid \mathcal{F}_t] - \nabla F_S(x_t)\|^2\big] + \sum_{t=0}^{T-1} \frac{L\eta^2}{2}\mathbb{E}\big[\|g_t\|^2\big].$$

By Lemma A.1 (a), it holds that

$$\mathbb{E}\big[\|g_t\|^2\big] = \mathbb{E}\big[\|\mathrm{clip}_\gamma(\nabla f(x_t; \xi_{i_t}))\|^2\big] \le \gamma^{2-p}\mathbb{E}\big[\|\nabla f(x_t; \xi_{i_t})\|^p\big] \le \gamma^{2-p}G^p.$$

Noting that $\nabla F_S(x_t) = \mathbb{E}[\nabla f(x_t; \xi_{i_t}) \mid \mathcal{F}_t]$, we have

$$\mathbb{E}[g_t \mid \mathcal{F}_t] - \nabla F_S(x_t) = \mathbb{E}\big[\mathrm{clip}_\gamma(\nabla f(x_t; \xi_{i_t})) - \nabla f(x_t; \xi_{i_t}) \mid \mathcal{F}_t\big].$$

Using $\|\mathbb{E}[X]\| \le \mathbb{E}[\|X\|]$ and Lemma A.1 (b), we obtain

$$\big\|\mathbb{E}[g_t \mid \mathcal{F}_t] - \nabla F_S(x_t)\big\| \le \mathbb{E}\big[\|\mathrm{clip}_\gamma(\nabla f(x_t; \xi_{i_t})) - \nabla f(x_t; \xi_{i_t})\| \mid \mathcal{F}_t\big]$$
$$\le \frac{1}{\gamma^{p-1}}\mathbb{E}\big[\|\nabla f(x_t; \xi_{i_t})\|^p \mid \mathcal{F}_t\big] \le \frac{G^p}{\gamma^{p-1}}.$$

Therefore, we have

$$\mathbb{E}\big[\|\mathbb{E}[g_t \mid \mathcal{F}_t] - \nabla F_S(x_t)\|^2\big] \le G^{2p}\gamma^{2-2p}.$$

Substituting the above bounds gives

$$\sum_{t=0}^{T-1} \frac{\eta}{2}\mathbb{E}\big[\|\nabla F_S(x_t)\|^2\big] \le \Delta + \sum_{t=0}^{T-1} \frac{\eta}{2}G^{2p}\gamma^{2-2p} + \sum_{t=0}^{T-1} \frac{L\eta^2}{2}\gamma^{2-p}G^p.$$

It follows that

$$\mathbb{E}_{S,A}\big[\|\nabla F_S(A(S))\|^2\big] \le \frac{2\Delta}{\eta T} + G^{2p}\gamma^{2-2p} + L\eta\gamma^{2-p}G^p.$$

By Jensen's inequality and the fact that $\sqrt{x+y+z} \le \sqrt{x} + \sqrt{y} + \sqrt{z}$ for $x, y, z \ge 0$, we have

$$\mathbb{E}_{S,A}\big[\|\nabla F_S(A(S))\|\big] \le \sqrt{\frac{2\Delta}{\eta T} + G^p \gamma^{1-p}} + \sqrt{L\eta G^p \gamma^{2-p}}.$$

Combining this bound with the generalization decomposition at the beginning completes the proof of Theorem 5.3.

### C.3. The Proof of Proposition 5.5

Without loss of generality, assume $S$ and $S'$ differ only at the $n$-th sample. By $L$-smoothness, for any $\xi$, one has

$$\mathbb{E}_A\Big[\big\|\nabla f(A(S);\xi) - \nabla f(A(S');\xi)\big\|^2\Big] \le L^2 \mathbb{E}_A\big[\|A(S) - A(S')\|^2\big].$$

For NSGD-B, the update is normalized, hence $\|x_{t+1} - x_t\| \le \eta_t$ and thus

$$\|A(S)\| \le \sum_{t=0}^{T-1} \eta_t \quad \text{and} \quad \|A(S')\| \le \sum_{t=0}^{T-1} \eta_t.$$

Let $I(A) = \{i_t^{(b)} : t = 0, \ldots, T-1, \; b = 1, \ldots, B\}$ be the sampled indices. It holds that $A(S) = A(S')$ on the event $\{n \notin I(A)\}$, and hence

$$\mathbb{E}_A\big[\|A(S) - A(S')\|^2\big] \le 4\Big(\sum_{t=0}^{T-1} \eta_t\Big)^2 \mathbb{P}(n \in I(A)) \le 4\Big(\sum_{t=0}^{T-1} \eta_t\Big)^2 \frac{BT}{n},$$

where we used $\mathbb{P}(n \in I(A)) \le \sum_{t=0}^{T-1} \sum_{b=1}^{B} \mathbb{P}(i_t^{(b)} = n) = BT/n$. Combining the above inequalities completes the proof.

### C.4. The Proof of Theorem 5.7

By Theorem 4.2 and Proposition 5.5, we have

$$\mathbb{E}_{S,A}\big[\|\nabla F(A(S))\|\big] \le \mathbb{E}_{S,A}\big[\|\nabla F_S(A(S))\|\big] + 8L\eta T\sqrt{\frac{BT}{n}} + C_p \sigma_p n^{-\frac{p-1}{p}}.$$

We next establish an upper bound on the optimization error of mini-batch normalized SGD. Let $\mathcal{F}_t$ be the sigma-field generated by $\{S, \{i_s^{(b)}\}_{0 \le s \le t-1, \, 1 \le b \le B}\}$. By $L$-smoothness of $F_S$ and the update rules, we have

$$F_S(x_{t+1}) \le F_S(x_t) + \big\langle \nabla F_S(x_t), x_{t+1} - x_t \big\rangle + \frac{L}{2}\|x_{t+1} - x_t\|^2$$
$$= F_S(x_t) - \eta\Big\langle \nabla F_S(x_t), \frac{g_t}{\|g_t\|} \Big\rangle + \frac{L\eta^2}{2},$$

where we set $g_t/\|g_t\| = 0$ when $g_t = 0$. Applying Lemma A.2 with $u = \nabla F_S(x_t)$ and $v = g_t$ yields

$$F_S(x_{t+1}) \le F_S(x_t) - \eta\|\nabla F_S(x_t)\| + 2\eta\|g_t - \nabla F_S(x_t)\| + \frac{L\eta^2}{2}.$$

Summing over $t = 0, \ldots, T-1$ and using $F_S(x_T) \ge F_S^*$ give

$$\sum_{t=0}^{T-1} \|\nabla F_S(x_t)\| \le \frac{F_S(x_0) - F_S^*}{\eta} + 2\sum_{t=0}^{T-1} \|g_t - \nabla F_S(x_t)\| + \frac{L\eta T}{2}.$$

Taking expectation and writing $\Delta := \mathbb{E}_S\big[F_S(x_0) - F_S^*\big]$ lead to

$$\frac{1}{T}\sum_{t=0}^{T-1} \mathbb{E}_{S,A}\big[\|\nabla F_S(x_t)\|\big] \le \frac{\Delta}{\eta T} + \frac{2}{T}\sum_{t=0}^{T-1} \mathbb{E}_{S,A}\big[\|g_t - \nabla F_S(x_t)\|\big] + \frac{L\eta}{2}. \tag{14}$$

It remains to bound $\mathbb{E}_{S,A}\big[\|g_t - \nabla F_S(x_t)\|\big]$. We define

$$X_b := \nabla f(x_t; \xi_{i_t^{(b)}}) - \nabla F_S(x_t), \qquad g_t - \nabla F_S(x_t) = \frac{1}{B}\sum_{b=1}^{B} X_b,$$

and $\mathcal{G}_{t,b} := \sigma(\mathcal{F}_t, i_t^{(1)}, \dots, i_t^{(b)})$. Then $\{X_b\}_{b=1}^{B}$ is a martingale difference sequence with respect to $\{\mathcal{G}_{t,b}\}_{b=0}^{B}$, since

$$\mathbb{E}\big[X_b \mid \mathcal{G}_{t,b-1}\big] = \mathbb{E}\big[\nabla f(x_t; \xi_{i_t^{(b)}}) \mid \mathcal{F}_t\big] - \nabla F_S(x_t) = \nabla F_S(x_t) - \nabla F_S(x_t) = 0.$$

Therefore, by Jensen's inequality and Lemma A.3, we have

$$\mathbb{E}\big[\|g_t - \nabla F_S(x_t)\| \mid \mathcal{F}_t\big] = \frac{1}{B}\mathbb{E}\Big[\Big\|\sum_{b=1}^{B} X_b\Big\| \;\Big|\; \mathcal{F}_t\Big]$$

$$\leq \frac{1}{B}\mathbb{E}\Big[\Big\|\sum_{b=1}^{B} X_b\Big\|^p \;\Big|\; \mathcal{F}_t\Big]^{1/p}$$

$$\leq \frac{1}{B}\Big(2\sum_{b=1}^{B}\mathbb{E}\big[\|X_b\|^p \mid \mathcal{F}_t\big]\Big)^{1/p}.$$

To bound $\mathbb{E}[\|X_b\|^p \mid \mathcal{F}_t]$, we use $\|a-b\|^p \leq 2^{p-1}(\|a\|^p + \|b\|^p)$ and $\|\mathbb{E}[Z \mid \mathcal{F}_t]\|^p \leq \mathbb{E}[\|Z\|^p \mid \mathcal{F}_t]$ to obtain

$$\mathbb{E}\big[\|X_b\|^p \mid \mathcal{F}_t\big] \leq 2^{p-1}\Big(\mathbb{E}\big[\|\nabla f(x_t; \xi_{i_t^{(b)}})\|^p \mid \mathcal{F}_t\big] + \|\nabla F_S(x_t)\|^p\Big)$$

$$= 2^{p-1}\Big(\mathbb{E}\big[\|\nabla f(x_t; \xi_{i_t^{(b)}})\|^p \mid \mathcal{F}_t\big] + \big\|\mathbb{E}[\nabla f(x_t; \xi_{i_t^{(b)}}) \mid \mathcal{F}_t]\big\|^p\Big)$$

$$\leq 2^{p-1}\Big(\mathbb{E}\big[\|\nabla f(x_t; \xi_{i_t^{(b)}})\|^p \mid \mathcal{F}_t\big] + \mathbb{E}\big[\|\nabla f(x_t; \xi_{i_t^{(b)}})\|^p \mid \mathcal{F}_t\big]\Big)$$

$$\leq 2^p G^p.$$

Substituting back yields

$$\mathbb{E}\big[\|g_t - \nabla F_S(x_t)\| \mid \mathcal{F}_t\big] \leq \frac{1}{B}\Big(2B \cdot 2^p G^p\Big)^{1/p} \leq 4GB^{-\frac{p-1}{p}}.$$

Taking expectation over $(S, A)$ gives

$$\mathbb{E}_{S,A}\big[\|g_t - \nabla F_S(x_t)\|\big] \leq 4GB^{-\frac{p-1}{p}}. \tag{15}$$

Substituting equation (15) into equation (14) yields

$$\mathbb{E}_{S,A}\big[\|\nabla F_S(A(S))\|\big] = \frac{1}{T}\sum_{t=0}^{T-1}\mathbb{E}_{S,A}\big[\|\nabla F_S(x_t)\|\big] \leq \frac{\Delta}{\eta T} + 8GB^{-\frac{p-1}{p}} + \frac{L\eta}{2}.$$

Combining this bound with the generalization decomposition at the beginning completes the proof of Theorem 5.7.

### C.5. The Proof of Proposition 5.12

Without loss of generality, assume $S$ and $S'$ differ only at the $n$-th sample. By $L$-smoothness, for any $\xi$, one has

$$\mathbb{E}_A\Big[\big\|\nabla f(A(S); \xi) - \nabla f(A(S'); \xi)\big\|^2\Big] \leq L^2 \mathbb{E}_A\big[\|A(S) - A(S')\|^2\big].$$

Let $\{x_t\}_{t=0}^{T}$ and $\{x_t'\}_{t=0}^{T}$ be the iterates generated on $S$ and $S'$, with $x_0 = x_0' = 0$. For NSGD-M, the update is normalized, hence $\|x_{t+1} - x_t\| \leq \eta_t$ and thus

$$\|A(S)\| \leq \sum_{t=0}^{T-1} \eta_t \quad \text{and} \quad \|A(S')\| \leq \sum_{t=0}^{T-1} \eta_t.$$

Let $I(A) = \{i_t\}_{t=0}^{T-1}$ be the index sequence sampled by the algorithm. It holds that $A(S) = A(S')$ on the event $\{n \notin I(A)\}$, and hence

$$\mathbb{E}_A\big[\|A(S) - A(S')\|^2\big] \leq 4\Big(\sum_{t=0}^{T-1} \eta_t\Big)^2 \mathbb{P}(n \in I(A)) \leq 4\Big(\sum_{t=0}^{T-1} \eta_t\Big)^2 \frac{T}{n},$$

where we used $\mathbb{P}(n \in I(A)) \leq \sum_{t=0}^{T-1} \mathbb{P}(i_t = n) = T/n$. Combining the above inequalities completes the proof.

### C.6. The Proof of Theorem 5.13

By Theorem 4.2 and Proposition 5.12, we have

$$\mathbb{E}_{S,A}\big[\|\nabla F(A(S))\|\big] \leq \mathbb{E}_{S,A}\big[\|\nabla F_S(A(S))\|\big] + 8L\eta T\sqrt{\frac{T}{n}} + C_p\sigma_p n^{-\frac{p-1}{p}}.$$

We next establish an upper bound on the optimization error of normalized SGD with momentum. Let $\mathcal{F}_t$ be the sigma-field generated by $\{S, i_0, \ldots, i_{t-1}\}$. By $L$-smoothness of $F_S$ and the update rules, we have

$$F_S(x_{t+1}) \leq F_S(x_t) + \big\langle \nabla F_S(x_t), x_{t+1} - x_t \big\rangle + \frac{L}{2}\|x_{t+1} - x_t\|^2$$
$$= F_S(x_t) - \eta\Big\langle \nabla F_S(x_t), \frac{m_t}{\|m_t\|} \Big\rangle + \frac{L\eta^2}{2},$$

Applying Lemma A.2 with $u = \nabla F_S(x_t)$ and $v = m_t$ yields

$$F_S(x_{t+1}) \leq F_S(x_t) - \eta\|\nabla F_S(x_t)\| + 2\eta\|m_t - \nabla F_S(x_t)\| + \frac{L\eta^2}{2}.$$

Summing over $t = 0, \ldots, T-1$, using $F_S(x_T) \geq F_S^*$, and taking $\mathbb{E}_{S,A}$ give

$$\frac{1}{T}\sum_{t=0}^{T-1} \mathbb{E}_{S,A}\big[\|\nabla F_S(x_t)\|\big] \leq \frac{\Delta}{\eta T} + \frac{2}{T}\sum_{t=0}^{T-1} \mathbb{E}_{S,A}\big[\|m_t - \nabla F_S(x_t)\|\big] + \frac{L\eta}{2}, \tag{16}$$

where $\Delta := \mathbb{E}_S[F_S(x_0) - F_S^*]$.

Define $\delta_t := m_t - \nabla F_S(x_t)$ and $\zeta_t := \nabla f(x_t; \xi_{i_t}) - \nabla F_S(x_t)$. Then $\mathbb{E}[\zeta_t \mid \mathcal{F}_t] = 0$, and by $\|a - b\|^p \leq 2^{p-1}(\|a\|^p + \|b\|^p)$, we have

$$\mathbb{E}\big[\|\zeta_t\|^p \mid \mathcal{F}_t\big] \leq 2^{p-1}\Big(\mathbb{E}\big[\|\nabla f(x_t; \xi_{i_t})\|^p \mid \mathcal{F}_t\big] + \|\nabla F_S(x_t)\|^p\Big) \leq 2^p G^p. \tag{17}$$

We next bound $\mathbb{E}_{S,A}[\|\delta_t\|]$. For $t \geq 1$, using $m_t = \beta m_{t-1} + (1-\beta)\nabla f(x_t; \xi_{i_t})$ gives

$$\delta_t = \beta\delta_{t-1} + \beta\big(\nabla F_S(x_{t-1}) - \nabla F_S(x_t)\big) + (1-\beta)\zeta_t.$$

For all $t \geq 1$, iterating the above equation from $\delta_0$ yields

$$\delta_t = \beta^t\delta_0 + (1-\beta)\sum_{\tau=1}^{t}\beta^{t-\tau}\zeta_\tau + \beta\sum_{\tau=1}^{t}\beta^{t-\tau}\big(\nabla F_S(x_{\tau-1}) - \nabla F_S(x_\tau)\big). \tag{18}$$

For $t = 0$, since $m_{-1} = 0$, we have

$$\delta_0 = m_0 - \nabla F_S(x_0) = (1-\beta)\zeta_0 - \beta\nabla F_S(x_0).$$

Using equation (17) at $t = 0$, Jensen's inequality, we obtain

$$\mathbb{E}_{S,A}\big[\|\delta_0\|\big] \leq (1-\beta)\mathbb{E}_{S,A}\big[\|\zeta_0\|\big] + \beta\mathbb{E}_{S,A}\big[\|\nabla F_S(x_0)\|\big]$$
$$\leq (1-\beta)\mathbb{E}_{S,A}\big[\|\zeta_0\|^p\big]^{1/p} + \beta G \leq (1-\beta) \cdot 2G + \beta G \leq 2G.$$

We now bound the two sums in equation (18). For the drift term, by $L$-smoothness and $\|x_\tau - x_{\tau-1}\| \leq \eta$, one has

$$\left\| \beta \sum_{\tau=1}^{t} \beta^{t-\tau} \left( \nabla F_S(x_{\tau-1}) - \nabla F_S(x_\tau) \right) \right\| \leq \beta \sum_{\tau=1}^{t} \beta^{t-\tau} \cdot L\|x_{\tau-1} - x_\tau\| \leq L\eta\beta \sum_{k=0}^{t-1} \beta^k \leq \frac{L\eta\beta}{1-\beta}.$$

For the noise term, define $Y_\tau := \beta^{t-\tau}\zeta_\tau$ for $\tau = 1, \ldots, t$. Since $\mathbb{E}[\zeta_\tau \mid \mathcal{F}_\tau] = 0$ and $\beta^{t-\tau}$ is deterministic, $\{Y_\tau\}_{\tau=1}^{t}$ is a martingale difference sequence with respect to the filtration $\{\mathcal{F}_\tau\}_{\tau=1}^{t}$. By Jensen's inequality, Lemma A.3, and equation (17), one has

$$\mathbb{E}_{S,A}\left[ \left\| \sum_{\tau=1}^{t} \beta^{t-\tau}\zeta_\tau \right\| \right] \leq \mathbb{E}_{S,A}\left[ \left\| \sum_{\tau=1}^{t} \beta^{t-\tau}\zeta_\tau \right\|^p \right]^{1/p}$$

$$\leq \left( 2 \sum_{\tau=1}^{t} \beta^{(t-\tau)p} \mathbb{E}_{S,A}\left[ \|\zeta_\tau\|^p \right] \right)^{1/p} \leq \left( 2 \sum_{\tau=1}^{t} \beta^{(t-\tau)p} \cdot 2^p G^p \right)^{1/p}$$

$$= 2^{1+\frac{1}{p}} G \left( \sum_{k=0}^{t-1} \beta^{kp} \right)^{1/p} \leq 4G(1-\beta^p)^{-1/p}.$$

For all $t \geq 1$, combining above results yields

$$\mathbb{E}_{S,A}\left[ \|\delta_t\| \right] \leq 2G\beta^t + 4G(1-\beta)(1-\beta^p)^{-1/p} + \frac{L\eta\beta}{1-\beta}.$$

Since $1 - \beta^p \geq 1 - \beta$ for $\beta \in [0,1)$ and $p \in (1,2]$, we have

$$(1-\beta)(1-\beta^p)^{-1/p} \leq (1-\beta)^{\frac{p-1}{p}}.$$

Thus, for all $t \geq 1$, one has

$$\mathbb{E}_{S,A}\left[ \|m_t - \nabla F_S(x_t)\| \right] \leq 2G\beta^t + 4G(1-\beta)^{\frac{p-1}{p}} + \frac{L\eta\beta}{1-\beta}, \tag{19}$$

and for $t = 0$, we have $\mathbb{E}_{S,A}[\|m_0 - \nabla F_S(x_0)\|] \leq 2G$.

Summing equation (19) over $t = 1, \ldots, T-1$ and using $\sum_{t=1}^{T-1} \beta^t \leq \frac{\beta}{1-\beta}$, we obtain

$$\sum_{t=0}^{T-1} \mathbb{E}_{S,A}\left[ \|m_t - \nabla F_S(x_t)\| \right] \leq 2G + \frac{2G\beta}{1-\beta} + 4G(1-\beta)^{\frac{p-1}{p}}T + \frac{L\eta\beta}{1-\beta}T. \tag{20}$$

Substituting equation (20) into equation (16) yields

$$\mathbb{E}_{S,A}\left[ \|\nabla F_S(A(S))\| \right] \leq \frac{\Delta}{\eta T} + \frac{4G}{T} + \frac{4G\beta}{(1-\beta)T} + 8G(1-\beta)^{\frac{p-1}{p}} + \frac{2L\eta\beta}{1-\beta} + \frac{L\eta}{2}.$$

Combining this bound with the generalization decomposition at the beginning completes the proof of Theorem 5.13.

### C.7. The Proof of Proposition 5.16

Without loss of generality, assume $S$ and $S'$ differ only at the $n$-th sample. By $L$-smoothness, for any $\xi$, one has

$$\mathbb{E}_A\left[ \left\| \nabla f(A(S); \xi) - \nabla f(A(S'); \xi) \right\|^2 \right] \leq L^2 \mathbb{E}_A\left[ \|A(S) - A(S')\|^2 \right].$$

Let $\{x_t\}_{t=0}^{T}$ and $\{x'_t\}_{t=0}^{T}$ be the iterates generated on $S$ and $S'$, with $x_0 = x'_0 = 0$. For NSGD-CM, the update is normalized, hence $\|x_{t+1} - x_t\| \leq \eta_t$ and thus

$$\|A(S)\| \leq \sum_{t=0}^{T-1} \eta_t, \quad \text{and} \quad \|A(S')\| \leq \sum_{t=0}^{T-1} \eta_t.$$

Let $I(A) = \{i_t\}_{t=0}^{T-1}$ be the index sequence sampled by the algorithm. It holds that $A(S) = A(S')$ on the event $\{n \notin I(A)\}$, and hence

$$\mathbb{E}_A\big[\|A(S) - A(S')\|^2\big] \leq 4\Big(\sum_{t=0}^{T-1} \eta_t\Big)^2 \mathbb{P}(n \in I(A)) \leq 4\Big(\sum_{t=0}^{T-1} \eta_t\Big)^2 \frac{T}{n},$$

where we used $\mathbb{P}(n \in I(A)) \leq \sum_{t=0}^{T-1} \mathbb{P}(i_t = n) = T/n$. Combining the above inequalities completes the proof.

### C.8. The Proof of Theorem 5.17

By Theorem 4.2 and Proposition 5.16, we have

$$\mathbb{E}_{S,A}\big[\|\nabla F(A(S))\|\big] \leq \mathbb{E}_{S,A}\big[\|\nabla F_S(A(S))\|\big] + 8L\eta T\sqrt{\frac{T}{n}} + C_p\sigma_p n^{-\frac{p-1}{p}}.$$

We next establish an upper bound on the optimization error of NSGD-CM. Let $\mathcal{F}_t$ be the sigma-field generated by $\{S, i_0, \ldots, i_{t-1}\}$. Recall that

$$g_t := \mathrm{clip}_\gamma\big(\nabla f(x_t; \xi_{i_t})\big), \qquad m_t := \beta m_{t-1} + (1-\beta)g_t, \qquad x_{t+1} = x_t - \eta\frac{m_t}{\|m_t\|},$$

and $m_{-1} = 0$. By $L$-smoothness of $F_S$, it holds that

$$F_S(x_{t+1}) \leq F_S(x_t) + \langle \nabla F_S(x_t), x_{t+1} - x_t\rangle + \frac{L}{2}\|x_{t+1} - x_t\|^2$$

$$= F_S(x_t) - \eta\Big\langle \nabla F_S(x_t), \frac{m_t}{\|m_t\|}\Big\rangle + \frac{L\eta^2}{2}$$

$$\leq F_S(x_t) - \eta\|\nabla F_S(x_t)\| + 2\eta\|m_t - \nabla F_S(x_t)\| + \frac{L\eta^2}{2},$$

where the last inequality uses Lemma A.2 with $u = \nabla F_S(x_t)$ and $v = m_t$. Summing over $t = 0, \ldots, T-1$, using $F_S(x_T) \geq F_S^*$, and taking $\mathbb{E}_{S,A}$ yield

$$\frac{1}{T}\sum_{t=0}^{T-1}\mathbb{E}_{S,A}\big[\|\nabla F_S(x_t)\|\big] \leq \frac{\Delta}{\eta T} + \frac{2}{T}\sum_{t=0}^{T-1}\mathbb{E}_{S,A}\big[\|m_t - \nabla F_S(x_t)\|\big] + \frac{L\eta}{2}, \tag{21}$$

where $\Delta := \mathbb{E}_S[F_S(x_0) - F_S^*]$.

Define $\delta_t := m_t - \nabla F_S(x_t)$. For $t \geq 1$, using $m_t = \beta m_{t-1} + (1-\beta)g_t$ gives

$$\delta_t = \beta\delta_{t-1} + \beta\big(\nabla F_S(x_{t-1}) - \nabla F_S(x_t)\big) + (1-\beta)\big(g_t - \nabla F_S(x_t)\big).$$

For all $t \geq 1$, iterating the above recursion from $\delta_0$ yields

$$\delta_t = \beta^t\delta_0 + (1-\beta)\sum_{\tau=1}^{t}\beta^{t-\tau}\big(g_\tau - \nabla F_S(x_\tau)\big) + \beta\sum_{\tau=1}^{t}\beta^{t-\tau}\big(\nabla F_S(x_{\tau-1}) - \nabla F_S(x_\tau)\big). \tag{22}$$

For $t = 0$, since $m_{-1} = 0$ and $m_0 = (1-\beta)g_0$, we have

$$\delta_0 = (1-\beta)g_0 - \nabla F_S(x_0).$$

Using $\|g_0\| \leq \|\nabla f(x_0; \xi_{i_0})\|$ and Jensen's inequality, we obtain

$$\mathbb{E}_{S,A}\big[\|\delta_0\|\big] \leq (1-\beta)\mathbb{E}_{S,A}\big[\|g_0\|\big] + \mathbb{E}_{S,A}\big[\|\nabla F_S(x_0)\|\big] \leq (1-\beta)G + G \leq 2G. \tag{23}$$

We now bound $\mathbb{E}_{S,A}\big[\|\delta_t\|\big]$ for $t \geq 1$ by controlling the two sums in equation (22). For the drift term, by $L$-smoothness and $\|x_\tau - x_{\tau-1}\| \leq \eta$, one has

$$\Big\|\beta\sum_{\tau=1}^{t}\beta^{t-\tau}\big(\nabla F_S(x_{\tau-1}) - \nabla F_S(x_\tau)\big)\Big\| \leq L\eta\beta\sum_{k=0}^{t-1}\beta^k \leq \frac{L\eta\beta}{1-\beta}.$$

For the noise term, we decompose

$$g_\tau - \nabla F_S(x_\tau) = \left(g_\tau - \mathbb{E}[g_\tau \mid \mathcal{F}_\tau]\right) + \left(\mathbb{E}[g_\tau \mid \mathcal{F}_\tau] - \nabla F_S(x_\tau)\right).$$

Then it holds

$$\mathbb{E}_{S,A}\left[\left\|\sum_{\tau=1}^{t}\beta^{t-\tau}\left(g_\tau - \nabla F_S(x_\tau)\right)\right\|\right]$$

$$\leq \mathbb{E}_{S,A}\left[\left\|\sum_{\tau=1}^{t}\beta^{t-\tau}\left(g_\tau - \mathbb{E}[g_\tau \mid \mathcal{F}_\tau]\right)\right\|\right] + \sum_{\tau=1}^{t}\beta^{t-\tau}\mathbb{E}_{S,A}\left[\left\|\mathbb{E}[g_\tau \mid \mathcal{F}_\tau] - \nabla F_S(x_\tau)\right\|\right].$$

For the martingale term, set $X_\tau := \beta^{t-\tau}\left(g_\tau - \mathbb{E}[g_\tau \mid \mathcal{F}_\tau]\right)$ for $\tau = 1, \ldots, t$. Then $\mathbb{E}[X_\tau \mid \mathcal{F}_\tau] = 0$ and $\{X_\tau\}_{\tau=1}^{t}$ is a martingale difference sequence. By Jensen's inequality and Lemma A.3, we have

$$\mathbb{E}_{S,A}\left[\left\|\sum_{\tau=1}^{t}\beta^{t-\tau}\left(g_\tau - \mathbb{E}[g_\tau \mid \mathcal{F}_\tau]\right)\right\|\right] \leq \mathbb{E}_{S,A}\left[\left\|\sum_{\tau=1}^{t}X_\tau\right\|^p\right]^{1/p} \leq \left(2\sum_{\tau=1}^{t}\mathbb{E}_{S,A}\left[\|X_\tau\|^p\right]\right)^{1/p}.$$

Moreover, by Jensen's inequality, we have

$$\mathbb{E}\left[\|g_\tau - \mathbb{E}[g_\tau \mid \mathcal{F}_\tau]\|^p \mid \mathcal{F}_\tau\right] \leq 2^p\mathbb{E}\left[\|g_\tau\|^p \mid \mathcal{F}_\tau\right] \leq 2^p\mathbb{E}\left[\|\nabla f(x_\tau; \xi_{i_\tau})\|^p \mid \mathcal{F}_\tau\right] \leq 2^pG^p,$$

and hence $\mathbb{E}_{S,A}[\|g_\tau - \mathbb{E}[g_\tau \mid \mathcal{F}_\tau]\|^p] \leq 2^pG^p$. Therefore, it holds that

$$\mathbb{E}_{S,A}\left[\left\|\sum_{\tau=1}^{t}\beta^{t-\tau}\left(g_\tau - \mathbb{E}[g_\tau \mid \mathcal{F}_\tau]\right)\right\|\right] \leq 2^{1+\frac{1}{p}}G\left(\sum_{\tau=1}^{t}\beta^{(t-\tau)p}\right)^{1/p} \leq 4G(1-\beta^p)^{-1/p}.$$

For the clipping bias term, using $\|\mathbb{E}[U]\| \leq \mathbb{E}[\|U\|]$ and Lemma A.1 (b), we obtain

$$\|\mathbb{E}[g_\tau \mid \mathcal{F}_\tau] - \nabla F_S(x_\tau)\| = \left\|\mathbb{E}\left[\mathrm{clip}_\gamma(\nabla f(x_\tau; \xi_{i_\tau})) - \nabla f(x_\tau; \xi_{i_\tau}) \mid \mathcal{F}_\tau\right]\right\|$$

$$\leq \mathbb{E}\left[\|\mathrm{clip}_\gamma(\nabla f(x_\tau; \xi_{i_\tau})) - \nabla f(x_\tau; \xi_{i_\tau})\| \mid \mathcal{F}_\tau\right]$$

$$\leq \gamma^{1-p}\mathbb{E}\left[\|\nabla f(x_\tau; \xi_{i_\tau})\|^p \mid \mathcal{F}_\tau\right]$$

$$\leq G^p\gamma^{1-p}.$$

Therefore, it holds that

$$\sum_{\tau=1}^{t}\beta^{t-\tau}\mathbb{E}_{S,A}\left[\|\mathbb{E}[g_\tau \mid \mathcal{F}_\tau] - \nabla F_S(x_\tau)\|\right] \leq G^p\gamma^{1-p}\sum_{\tau=1}^{t}\beta^{t-\tau} \leq \frac{G^p\gamma^{1-p}}{1-\beta}.$$

Combining the two bounds and multiplying by $(1-\beta)$, we obtain

$$(1-\beta)\mathbb{E}_{S,A}\left[\left\|\sum_{\tau=1}^{t}\beta^{t-\tau}\left(g_\tau - \nabla F_S(x_\tau)\right)\right\|\right] \leq 4G(1-\beta)(1-\beta^p)^{-1/p} + G^p\gamma^{1-p}.$$

Since $1 - \beta^p \geq 1 - \beta$ for $\beta \in [0,1)$ and $p \in (1,2]$, it holds that

$$(1-\beta)(1-\beta^p)^{-1/p} \leq (1-\beta)^{\frac{p-1}{p}}.$$

Substituting the above bounds into equation (22) and using equation (23) yields

$$\mathbb{E}_{S,A}\left[\|\delta_t\|\right] \leq 2G\beta^t + 4G(1-\beta)^{\frac{p-1}{p}} + G^p\gamma^{1-p} + \frac{L\eta\beta}{1-\beta}, \tag{24}$$

for all $t \geq 1$ and for $t = 0$, equation (23) gives $\mathbb{E}_{S,A}[\|\delta_0\|] \leq 2G$.

Summing equation (24) over $t = 1, \dots, T-1$ and using $\sum_{t=1}^{T-1} \beta^t \leq \frac{\beta}{1-\beta}$, we obtain

$$\sum_{t=0}^{T-1} \mathbb{E}_{S,A}\big[\|m_t - \nabla F_S(x_t)\|\big] \leq 2G + \frac{2G\beta}{1-\beta} + 4G(1-\beta)^{\frac{p-1}{p}}T + G^p\gamma^{1-p}T + \frac{L\eta\beta}{1-\beta}T. \tag{25}$$

Substituting equation (25) into equation (21) yields

$$\mathbb{E}_{S,A}\big[\|\nabla F_S(A(S))\|\big] \leq \frac{\Delta}{\eta T} + \frac{4G}{T} + \frac{4G\beta}{(1-\beta)T} + 8G(1-\beta)^{\frac{p-1}{p}} + 2G^p\gamma^{1-p} + \frac{2L\eta\beta}{1-\beta} + \frac{L\eta}{2}.$$

Combining this bound with the generalization decomposition at the beginning completes the proof of Theorem 5.17.

## D. Discussion of Mini-Batch Variants

In this section, we discuss how the mini-batch argument used in the proof of Theorem 5.7 can be adapted to the other algorithms in Section 5. We take the mini-batch version of NSGD-M as an example. At iteration $t$, sample $i_t^{(1)}, \dots, i_t^{(B)}$ independently and uniformly from $[n]$, and the mini-batch NSGD-M iterates as

$$\bar{g}_t = \frac{1}{B}\sum_{b=1}^{B} \nabla f(x_t; \xi_{i_t^{(b)}}), \quad m_t = \beta m_{t-1} + (1-\beta)\bar{g}_t, \quad \text{and} \quad x_{t+1} = x_t - \eta\frac{m_t}{\|m_t\|}.$$

The stability analysis is the same as that of Proposition 5.12, except that the probability of querying the replaced sample is bounded by $BT/n$. Hence the mini-batch NSGD-M is $\epsilon_{m,B}$-uniformly stable in gradients with

$$\epsilon_{m,B} \leq 2L\sqrt{\frac{BT}{n}}\sum_{t=0}^{T-1}\eta_t.$$

It remains to identify the change in the optimization analysis. Let

$$\zeta_t = \bar{g}_t - \nabla F_S(x_t) = \frac{1}{B}\sum_{b=1}^{B} X_{t,b}, \quad \text{and} \quad X_{t,b} = \nabla f(x_t; \xi_{i_t^{(b)}}) - \nabla F_S(x_t).$$

Conditioned on the history before iteration $t$, the sequence $\{X_{t,b}\}_{b=1}^{B}$ is a martingale difference sequence. By Lemma A.3 and the assumption $\mathbb{E}_{i_t^{(b)}}[\|\nabla f(x_t; \xi_{i_t^{(b)}})\|^p] \leq G^p$, we have

$$\begin{aligned}
\mathbb{E}[\|\zeta_t\|^p \mid \mathcal{F}_t] &= \frac{1}{B^p}\mathbb{E}\left[\left\|\sum_{b=1}^{B} X_{t,b}\right\|^p \middle| \mathcal{F}_t\right] \\
&\leq \frac{2}{B^p}\sum_{b=1}^{B}\mathbb{E}[\|X_{t,b}\|^p \mid \mathcal{F}_t] \\
&\leq 8G^p B^{1-p}.
\end{aligned}$$

Following the proof in Appendix C.6 with this bound gives

$$\begin{aligned}
\mathbb{E}_{S,A}[\|\nabla F(A(S))\|] \leq {} & \frac{\Delta}{\eta T} + \frac{L\eta}{2} + 8GB^{-\frac{p-1}{p}}(1-\beta)^{\frac{p-1}{p}} + \frac{4G}{T} + \frac{4G\beta}{(1-\beta)T} + \frac{2L\eta\beta}{1-\beta} \\
& + 8L\eta T\sqrt{\frac{BT}{n}} + C_p\sigma_p n^{-\frac{p-1}{p}}.
\end{aligned}$$

When $B = 1$, this bound reduces to the bound for NSGD-M in Theorem 5.13. The mini-batch variants of clipped SGD and NSGD-CM can be analyzed analogously.

