# OpenReview forum: "Stability and Generalization of Nonconvex Optimization with Heavy-Tailed Noise"
_ICML.cc/2026/Conference — ICML 2026 regular_

### Official Review · Reviewer_zKko · 2026-03-08

**Soundness:** 3
**Presentation:** 3
**Significance:** 3
**Originality:** 2
**Overall Recommendation:** 4
**Confidence:** 4

**Summary:**

The authors study generalization analysis for stochastic nonconvex optimization under heavy-tailed gradient noise. They investigate whether stability-based generalization guarantees—commonly established under bounded variance assumptions—can be extended to the more realistic heavy-tailed regime, where the noise in gradient estimation only has bounded $p$-th moment ($p$-BCM condition). The central technical contribution is a truncation and tail decomposition argument that extends the stability-based bound of Lei (2023) to the p-BCM case. This framework is then applied to several stochastic optimization algorithms—clipped SGD, mini-batch normalized SGD, normalized SGD with momentum, and normalized SGD with clipping and momentum.

**Compliance With Llm Reviewing Policy:**

Affirmed.

**Key Questions For Authors:**

1. Dependence of bounds on algorithm parameters.
Several results involve multiple hyperparameters (e.g., clipping threshold, momentum, step size, iteration count). How sensitive are the derived generalization bounds to these parameters?
2. In Liu & Zhou (2025), the authors also investigated the cases where algorithm parameters are unknown. Can the results in this paper be extended to the unknown parameter cases as well?
3. The proposed framework analyzes several variants of SGD. Could the authors (briefly) discuss whether their approach could also extend to adaptive optimizers (e.g., Adam-like methods) or other stochastic optimization algorithms?

**Limitations:**

yes

**Strengths And Weaknesses:**

Strengths:

1. Heavy-tailed gradient noise is widely observed in deep learning and RL. Extending generalization analysis to this regime is timely.
2. The truncation argument combined with stability decomposition is conceptually clean and standard in heavy-tail analysis but applied here in a new context.
3. The framework is applied to multiple variants of SGD, which helps illustrate the utility of the general bound.
4. When $p=2$, the bound recovers to the bounded variance result of Lei (2023).

Weaknesses:
1. This is not a weakness, but I write it here for clarity. After the submission of this paper, there have been some more papers analyzing heavy-tailed noise of SGD and its variants, e.g., "Why is Normalization Preferred? A Worst-Case Complexity Theory for Stochastically Preconditioned SGD under Heavy-Tailed Noise" (arXiv:2602.13413), and "Complexity of normalized stochastic first-order methods with momentum under heavy-tailed noise"  (arXiv:2506.11214), and "Sign-Based Optimizers Are Effective Under Heavy-Tailed Noise" (arXiv:2506.11214), and so on. This is, of course, not a weakness of the paper, but please update the literature review to make it up to date.
2. The sharpness of the generalization bound should be better discussed. The bound is proportional to $n^{-(p-1)/p}$, which degrades significantly when $p$ goes to 1. It is unclear if the result is tight or loose.
3. No experiment is conducted for this paper. Simulations can enhance the readability and credibility of this paper.
4. Several assumptions are relatively strong and nonstandard:
4.1: Many results assume $x_0=0$. This is weird for proving a result that is supposed to hold globally, where arbitrary initialization should be allowed. The authors should deal with this problem.
4.2: The analysis additionally assumes bounded moment of gradient norm $\mathbb{E}[\|\nabla f(x;\xi)\|^p] \leq G^p$. Such an assumption is not adopted by recent literature, e.g.,  Liu & Zhou (2025), Hubler et al. (2025). The authors might justify the necessity of this assumption.
5. Lack of broader theoretical implications: The paper focuses narrowly on extending a single stability bound, but its coverage can be significantly enhanced if the authors can discuss its implications for learning theory and connections to robust statistics.

---

> ### Author Rebuttal · Authors · 2026-03-30
>
> Thank you for your careful reading and valuable comments.
>
> **Weaknesses:**
>
> > please update the literature review to make it up to date.
>
> **A1:** Thank you for pointing out these related works. We are happy to cite these references in the literature review of the revision.
>
> > The sharpness of the generalization bound should be better discussed.
>
> **A2:** Thank you for your insightful question, which is also raised by Reviewer WNyw. Please refer to A5 to Reviewer WNyw for details.
>
> > No experiment is conducted for this paper.
>
> **A3:** The goal of this paper is to establish  generalization guarantees for existing optimization algorithms for heavy-tailed noise, rather than proposing new algorithms. For the clipping and normalization methods studied in this paper, numerous empirical studies (e.g., [1, 2, 3]) have validated their advantages in practice compared to vanilla SGD.
>
> > Several assumptions are relatively strong and nonstandard.
>
> **A4:** Thank you for your insightful comments.
> 1. The assumption of $x_0=0$ is only for the ease of presentation, which is common in the analysis of stability. **Our main results still hold without the assumption $x_0=0$**. Please refer to A2 to Reviewer zGEd for details.
>
> 2. The assumption of $\mathbb{E}\bigl[\\|\nabla f(x_t;\xi_t)\\|^{p}\bigr]\le G^{p}$ is commonly used in the algorithmic stability-based generalization analysis [4, 5]. For normalized SGD and its variants, this assumption **can be relaxed to a weaker condition** as follows
> $$
> \mathbb{E}\_S \Big[ \frac{1}{n}\sum\_{i=1}^n \bigl\\| \nabla f(x\_t(S);\xi_i)-\nabla F\_S(x\_t(S)) \bigr\\|^p \Big] \le \sigma_p^p.\tag{1}
> $$
> Please refer to A3 to Reviewer WNyw for details. In Liu \& Zhou (2025) and Hubler et al. (2025), they study the optimization error in the online setting rather than the generalization error, thus do not require the assumption used for the generalization analysis in our paper.
>
> > Lack of broader theoretical implications
>
> **A5:** We emphasize that **our paper develops a general framework for analyzing the generalization bound in heavy-tailed setting**, rather than a single stability bound. In particular, our Theorem 4.2 can be used to analyze many popular optimization algorithms for heavy-tailed noise, leading to generalization bounds of Clipped SGD, NSGD-B, NSGD-M, and NSGD-CM in Theorems 5.2, 5.5, 5.9 and 5.13, respectively. Therefore, we believe our results have clear broader theoretical implications in statistical learning theory.
>
> **Questions:**
>
> > Dependence of bounds on algorithm parameters.
>
> **A6:** As we mention in item 2 of A1 to Reviewer tHY9, the parameter settings derived by minimizing the population gradient bound align with those in the work on designing optimization algorithms [3]. Therefore, **the sensitivity to the parameter setting in the generalization bounds follows that in the work focusing on analyzing optimization error** [3].
>
> >... extended to the unknown parameter cases ...
>
> **A7:** The results in our paper can also be extended to the unknown-parameter case. Combining the decomposition on the population gradient norm in line 179 and generalization error bound (Theorem 4.2), we achieve
> $$
> \mathbb{E}\_{S,A}\bigl[\\|\nabla F(A(S))\\|\bigr] \le \underbrace{\mathbb{E}_{S,A} \left[ \left\\| \nabla F_S(A(S)) \right\\| \right]}\_{\text{optimization error}} + 4\epsilon + C_p\sigma_p n^{-\frac{p-1}{p}}, \tag{2}
> $$
> which **does not depend on specific optimization algorithm**.
> By incorporating the optimization error obtained by Liu and Zhou [Theorem 3.6, 6] into above upper bound, we can obtain the population gradient bound under the unknown-parameter case.
>
> >... extend to adaptive optimizers ...
>
> **A8:** As we mentioned in A7, the upper bound in Eq. (2) does not depend on specific optimization algorithm.
> Therefore, we can directly establish the population gradient bound by combining Eq. (2) with the existing optimization error bound of adaptive optimizers [7].
>
> ---
> References
>
> [1] Jingzhao Zhang et al. Why are adaptive methods good for attention models? In NeurIPS 2020.
>
> [2] Xingyu Wang et al. Eliminating sharp minima from SGD with truncated heavy-tailed noise. In ICLR 2022.
>
> [3] Florian Hübler et al. From gradient clipping to normalization for heavy tailed SGD. In AISTATS 2025.
>
> [4] Yunwen Lei. Stability and generalization of stochastic optimization with nonconvex and nonsmooth problems. In COLT 2023.
>
> [5] Moritz Hardt et al. Train faster, generalize better: stability of stochastic gradient descent. In ICML 2016.
>
> [6] Zijian Liu and Zhengyuan Zhou. Nonconvex stochastic optimization under heavy-tailed noises: optimal convergence without gradient clipping. In ICLR 2025.
>
> [7] Savelii Chezhegov et al. Clipping improves Adam-norm and AdaGrad-norm when the noise is heavy-tailed. In ICML 2025.

---

> > ### Author Rebuttal · Reviewer_zKko · 2026-03-31
> >
> > I think the rebuttal has reasonably resolved my questions. I have no further questions for the authors. If the final version of the paper can be revised as described in the rebuttal, I support the publication of this paper.

---

> > > ### Author Response · Authors · 2026-04-03
> > >
> > > Thank you for your reply and support. We will revise the paper accordingly and further improve the final version as described in the rebuttal.

---

### Official Review · Reviewer_tHY9 · 2026-03-10

**Soundness:** 3
**Presentation:** 3
**Significance:** 3
**Originality:** 3
**Overall Recommendation:** 4
**Confidence:** 4

**Summary:**

For nonconvex problems satisfying the p-BCM condition, this paper leverages the technique of truncation argument to establish the connection between algorithmic stability and generalization. Since the restrictive bounded variance assumption is a special case of the p-BCM condition when $p=2$, results of this paper extend the applicability of stability-based generalization analysis from both theoretical and practical perspectives. This connection is then applied to clipped and normalized SGD, as well as their mini-batch and momentum variants, yielding comprehensive generalization guarantees. The authors present both stability bounds and population gradient norm bounds for each specific algorithm. Furthermore, optimal convergence rates can be achieved by tuning the algorithmic parameters and employing the early stopping strategy.

**Compliance With Llm Reviewing Policy:**

Affirmed.

**Final Justification:**

The rebuttal has addressed my concerns regarding the assumption and the potential extension. The technical contribution of this paper is significant, which lies in introducing the truncation argument to extend the bounded variance assumption to the p-BCM condition in the nonconvex setting. I recommend accepting this paper.

**Key Questions For Authors:**

1. Are there significant differences between the bounded raw moment condition [R1] and the p-BCM condition in terms of theoretical analysis? How are they directly related to the mathematical formulation of heavy-tailed distributions?
2. Without the smoothness assumption, is it possible to use the truncation argument to establish stability-based generalization bounds for Moreau envelope under the p-BCM condition?

[R1] Zhang J, Karimireddy S P, Veit A, et al. Why ADAM beats SGD for attention models, 2019.

**Limitations:**

Yes

**Strengths And Weaknesses:**

Strengths
1. The technical contribution of this paper lies in introducing the truncation argument, which extends the bounded variance assumption to the p-BCM condition for stability-based generalization under the nonconvex setting.
2. The authors apply the established connection to various variants of SGD, providing theoretical insights into a broad range of nonconvex optimization tasks with heavy-tailed noise.
3. The authors demonstrate a comprehensive understanding of prior work. The paper is technically rigorous and well supported by extensive empirical evidence.

Weaknesses
1. In Section 5, the remarks on the theoretical results could be strengthened. For example, comparing the established results with those derived under the bounded variance assumption could further highlight the theoretical contributions of this paper. Furthermore, it is worth noting how theoretical results can provide valuable insights to guide the setting of algorithmic parameters in practical applications.

---

> ### Author Rebuttal · Authors · 2026-03-30
>
> Thank you for your constructive comments and valuable suggestions.
>
> **Weaknesses:**
>
> > In Section 5, the remarks on the theoretical results could be strengthened. For example, comparing the established results with those derived under the bounded variance assumption could further highlight the theoretical contributions of this paper. Furthermore, it is worth noting how theoretical results can provide valuable insights to guide the setting of algorithmic parameters in practical applications.
>
> **A1:** Thanks for your insightful comments. We address your questions as follows.
>
> 1. To the best of our knowledge, the most related work to our Section 5 is the analysis of NSGD-M by Sun et al. [1].
> Specifically, Sun et al. [1] provided a generalization bound of $O(n^{-1/8})$ for NSGD-M under the assumption of bounded variance, which exactly **matches the result of our Corollary 5.10 in the case $p=2$**.
>
> 2. We have provided the guidance for the algorithmic parameters setting in Corollaries 5.3, 5.6, 5.10, and 5.14. For example, Theorem 5.5 shows that mini-batch normalized SGD has the population gradient bound of
> $$
> \mathbb{E}\_{S,A}\bigl[\\|\nabla F(A(S))\\|\bigr] \le \frac{\Delta}{\eta T} + \frac{L\eta}{2} + 4GB^{-\frac{p-1}{p}} + 8L \eta T\sqrt{\frac{BT}{n}} + C\_p\sigma\_p n^{-\frac{p-1}{p}}.
> $$
> We can minimize the upper bound in Eq. (1) by taking $$T \asymp n^{\frac{2(p-1)}{7p-6}},\quad \eta \asymp T^{-\frac{1}{2}} \asymp n^{-\frac{p-1}{7p-6}},\quad \text{and} \quad B \asymp T^{\frac{p}{2p-2}} \asymp n^{\frac{p}{7p-6}},$$
> which leads to the population gradient bound of $O\Bigl(n^{-\frac{p-1}{7p-6}}\Bigr)$.
> The above settings of $\eta \asymp T^{-\frac{1}{2}}$ and $B \asymp T^{\frac{p}{2p-2}}$ **match those derived from the optimization perspective** (Corollary 3 of Hübler et al. [2]).
>
> We will include this discussion in the revision.
>
> **Questions:**
>
> > Are there significant differences between the bounded raw moment condition [R1] and the p-BCM condition in terms of theoretical analysis? How are they directly related to the mathematical formulation of heavy-tailed distributions?
>
> > [R1] Zhang J, Karimireddy S P, Veit A, et al. Why ADAM beats SGD for attention models, 2019.
>
> **A2:** The bounded raw moment condition in Zhang et al. [R1] is
> \begin{align*}
> \mathbb{E}\bigl[\\|\nabla f(x;\xi)\\|^p\bigr] \le \sigma\_p^p,
> \end{align*}
> **which is stronger than the $p$-BCM condition used in our analysis**.
> Indeed, by the triangle inequality and Jensen's inequality, it holds
> $$
> \mathbb{E}\Bigl[\bigl\\|\nabla f(x;\xi)-\nabla F(x)\bigr\\|^p\Bigr] \le 2^{p-1}\Bigl(\mathbb{E}\bigl[\\|\nabla f(x;\xi)\\|^p\bigr] + \\|\nabla F(x)\\|^p\Bigr) \le 2^p \mathbb{E}\bigl[\\|\nabla f(x;\xi)\\|^p\bigr] \le 4\sigma\_p^p.
> $$
>
> As discussed in lines 195--206, the $p$-BCM condition only assumes that the centered moment is upper bounded and it does not rely on the specific distribution.
> In particular, **many popular heavy-tailed distributions** including $\alpha$-stable, Student-$t$, and Pareto distributions **satisfy the $p$-BCM condition**.
>
> > Without the smoothness assumption, is it possible to use the truncation argument to establish stability-based generalization bounds for Moreau envelope under the $p$-BCM condition?
>
> **A3:** Our truncation argument is a general idea that decomposes the heavy-tailed noise into a truncated part and a residual tail (e.g., lines 246--263), which **does not depend on the smoothness**. For the nonsmooth case, we think it is possible to incorporate the optimization error derived from the Moreau envelope [3] into our framework to establish generalization bounds.
> We believe this is an interesting future direction and we will include the discussion in our revision.
>
> ---
> References
>
> [1] Tao Sun, Li Shen, and Xinwang Liu. On nonconvex SGD under unbounded noise with weak gradient lipschitz and delayed stochastic gradient. IEEE Transactions on Pattern Analysis and Machine Intelligence, 2025.
>
> [2] Florian Hübler, Ilyas Fatkhullin, and Niao He. From gradient clipping to normalization for heavy tailed SGD. In AISTATS 2025.
>
> [3] Tianxi Zhu, Yi Xu, and Xiangyang Ji. Stochastic weakly convex optimization under heavy-tailed noises. arXiv:2507.13283, 2025.

---

> > ### Author Rebuttal · Reviewer_tHY9 · 2026-04-01
> >
> > Thank you for the clarification. The rebuttal has successfully addressed my concerns. The inclusion of these points in the revised version will further strengthen the paper. I will maintain my score.

---

> > > ### Author Response · Authors · 2026-04-03
> > >
> > > Thank you for your reply. We will make the corresponding revisions to the paper based on the discussion.

---

### Official Review · Reviewer_WNyw · 2026-03-11

**Soundness:** 4
**Presentation:** 4
**Significance:** 3
**Originality:** 4
**Overall Recommendation:** 5
**Confidence:** 4

**Summary:**

The paper extends the stability and generalization of the non-convex optimization from the variance-bounded gradients to the heavy-tailed cases, with the $p$-th bounded center moment ($p$-BCM) assumed for $p\in(1,2]$. This case is in fact more commonly seen in the stochastic gradient descent. The primary proposed novel technique is a truncation-and-trail decomposition of the stochastic gradient noise, which helps provide different types of the error bounds. The section of the applications includes clipped SGD, mini-batch normalized SGD, normalized SGD with momentum, and normalized SGD with clipping plus momentum.

**Compliance With Llm Reviewing Policy:**

Affirmed.

**Final Justification:**

The paper is very interesting and is mathematically solid. The authors carefully addressed all my concerns in the rebuttal, which makes the novelty very clear. The restrictions are reasonable and I can see potential extensions in the future works in this area. Therefore, I decide to give a final score of 5 - Accept to support its publication.

**Key Questions For Authors:**

(1) For the normalized method, the stability bounds seem to come mainly from the fact that normalized updates have bounded length $\\|x_{t+1}-x_t\\|\le \eta_t$ (Please correct me if I were wrong), together with the probability that the replaced sample is used. Does this mean that the heavy-tail assumption enter only later through the generic $C_p\sigma_pn^{-(p-1)/p}$ on line 926-927 and the optimization-error part, instead of the stability analysis? It's not wrong, but the authors might want to make this clear when they phrase it in the main text;

(2) How sharp is the term $n^{-(p-1)/p}$ in Theorem 4.2? Could the authors derive a lower bound or at least discussing how optimal this rate is?

(3) Is there a particular reason why many places assumed $x_0=0$? For example, in Proposition 5.1 and Theorem 5.2. I didn't see how it is necessary in the proofs. Maybe the authors could add a one-sentence clarification?

**Limitations:**

Yes.

**Strengths And Weaknesses:**

Strengths:

(1) The theoretical contribution is significant. The authors first time bridged the gap between algorithmic stability and generalization error under the $p$-BCM assumptions, $p\in(1,2]$, which coincides with the previous literature for the special case $p=2$;

(2) The application part is very fruitful, covering multiple popular and efficient algorithms. The heavy-tailed setting prevents the usual second-moment expansion. However, the proposed truncation-plus-centering decomposition is quite novel and helpful to solve this issue;

(3) Beyond stating the bounds, the paper also draws a useful comparison between clipping and normalization, suggesting that normalized variants can achieve sharper population-gradient rates than clipped SGD.

Weaknesses:

(1) In the literature for simpler settings of the optimization under heavy tails, high-probability control is usually derived. The current paper is focused on the guarantees in expectation only. It will largely benefit the scope if the authors could discuss the probabilistic bounds;

(2) The paper studies the generalization error $\mathbb{E}\\|\nabla F(A(S)) - \nabla F_S(A(S))\\|$ and then bounds $\mathbb{E}\\|\nabla F(A(S))\\|$ by optimization error plus the gradient generalization error, which is good, but could have a stronger result by excessing population risk directly.

(3) I noticed that the paper assumes a uniform bound for all the iterates of the form $\mathbb{E}\\|\nabla f(x_t,\xi_t)\\|^p\le G^p$ for all $t$ (e.g., Theorem 5.2) and its mini-batch analogues. This is somewhat stronger than merely assuming heavy-tailed centered noise around $\nabla F(x)$. It would be helpful to justify this assumption by giving some examples or at least discussing the potential possibility to relax this assumption.

---

> ### Author Rebuttal · Authors · 2026-03-30
>
> Thank you for your careful reading and insightful comments.
>
> **Weaknesses:**
>
> > It will largely benefit the scope if the authors could discuss the probabilistic bounds.
>
> **A1:** Thank you for your valuable suggestion. The current analysis of the algorithmic stability-based generalization bounds relies on introducing ghost sample $\xi'$ (line 202) to handle the dependence between $A(S)$ and $S$. It remains unclear whether this technique can be extended to the high-probability setting, which is an interesting direction for future work.
>
> > but could have a stronger result by excessing population risk directly.
>
> **A2:** Thank you for your suggestion. It is worth noting that $A(S)$ is not an accurate solution (stationary point) to $\min_{x\in \mathbb{R}^d} F_S(x)$, but only an approximate solution (stationary point) returned by an algorithm. Thus, **there exists optimization error in the solution** and it is natural to bound the population risk $\mathbb{E}\\|\nabla F(A(S))\\|$ by optimization error plus the gradient generalization error.
>
> > It would be helpful to justify this assumption by giving some examples or at least discussing the potential possibility to relax this assumption.
>
> **A3:** The assumption of $\\mathbb{E}\\bigl[\\|\\nabla f(x\_t;\\xi\_t)\\|^{p}\\bigr]\\le G^{p}$ is commonly used in related works [1, 2, 3]. **For normalized SGD and its variants, this assumption can be relaxed** to a weaker condition as follows
> $$
> \\mathbb{E}\_S \\Big[ \\frac{1}{n}\\sum\_{i=1}^n \\bigl\\| \\nabla f(x\_t(S);\\xi\_i)-\\nabla F\_S(x\_t(S)) \\bigr\\|^p \\Big] \\le \\sigma\_p^p.\tag{1}
> $$
> Under condition (1), the gradient estimation error is upper bounded by
> \begin{align*}
> & \mathbb{E}\_{S,A}\Biggl[\Bigg\\|\frac{1}{B}\sum\_{b=1}^B\Bigl(\nabla f(x\_t(S);\xi\_{i\_t^{(b)}})-\nabla F\_S(x\_t(S))\Bigr) \Bigg\\|\Biggr]\\\\
> &=\mathbb{E}\_S\Biggl[\mathbb{E}\_A\Biggl[\Biggl\\|\frac{1}{B}\sum\_{b=1}^B\Bigl(\nabla f(x\_t(S);\xi\_{i\_t^{(b)}})-\nabla F\_S(x\_t(S))\Bigr)\Biggr\\|\Biggm|S\Biggr]\Biggr] \\\\
> &\le\frac{1}{B}\mathbb{E}\_S\Biggl[\mathbb{E}\_A\Biggl[\Biggl\\|\sum\_{b=1}^B\Bigl(\nabla f(x\_t(S);\xi\_{i\_t^{(b)}})-\nabla F\_S(x\_t(S))\Bigr)\\Biggr\\|^p\Biggm|S\Biggr]^{1/p}\Biggr] \\\\
> &\le\frac{2}{B^{(p-1)/p}}
> \Biggl(\mathbb{E}\_S\Biggl[\frac{1}{n}\\sum\_{i=1}^n\bigl\\|\nabla f(x\_t(S);\\xi\_i)-\nabla F\_S(x\_t(S))\\bigr\\|^p\\Biggr]\\Biggr)^{1/p} \\\\
> &\le\frac{2\\sigma\_p}{B^{(p-1)/p}}\tag{2},
> \end{align*}
> where the first inequality holds by Jensen's inequality; the second inequality follows from Lemma A.3; and the final inequality is based on the relaxed assumption (1).
> For the example of mini-batch normalized SGD, we can substitute (2) into the inequality in lines 1098, then the result of Theorem 5.5 still holds (up to a constant difference). The results of normalized SGD with momentum and normalized SGD with clipping and momentum also can be achieved similarly.
>
> Note that condition (1) cannot be directly achieved from the $p$-BCM assumption since $x\_t(S)$ and $\\xi\_i$ may not be independent.
>
> **Questions:**
> > Does this mean that the heavy-tail assumption enter only later through the generic $C_p \sigma_p n^{-(p-1)/p}$ on line 926-927 and the optimization-error part, instead of the stability analysis?
>
> **A4:** Thank you for your insightful observation. For normalized SGD, heavy-tailed noise indeed does not affect the stability, **which is exactly its advantage**. In contrast, the vanilla SGD $x_{t+1}=x_t-\eta g_t$ leads to $\\|x_{t+1}-x_t\\|=\eta_t\\|g_t\\|$, where $\\|g_t\\|$ can be extremely large under heavy-tailed noise that heavily affects the algorithmic stability.
> We will clarify this point in our revision.
>
> > How sharp is the term $n^{-(p-1)/p}$ in Theorem 4.2?
>
> **A5:** To the best of our knowledge, it is difficult to derive a lower bound for the algorithmic stability-based generalization result of nonconvex problems. On the other hand, Devroye et al. [4] indicate for empirical mean estimator, there exists a distribution with bounded $p$-th central moment such that the estimation error is at least of order $\Omega(n^{-(p-1)/p})$, **which may suggest the term $n^{-(p-1)/p}$ is optimal**. We are happy to include this discussion in our revision.
>
> > Is there a particular reason why many places assumed $x_0 = 0$?
>
> **A6:** The assumption of $x_0=0$ is only for the ease of presentation. **Our main results still hold without the assumption.** Please refer to A2 to Reviewer zGEd for details.
>
> ---
> References
>
> [1] Ashok Cutkosky and Harsh Mehta. High-probability bounds for non-convex stochastic optimization with heavy tails. In NeurIPS 2021.
>
> [2] Yunwen Lei. Stability and generalization of stochastic optimization with nonconvex and nonsmooth problems. In COLT 2023.
>
> [3] Moritz Hardt et al. Train faster, generalize better: stability of stochastic gradient descent. In ICML 2016.
>
> [4] Luc Devroye et al. Sub-Gaussian mean estimators. Annals of Statistics, 2016.

---

> > ### Author Rebuttal · Reviewer_WNyw · 2026-03-31
> >
> > Thank you very much for addressing my concerns and clarifying some of my confusions. It would be great to see these proposed revisions added to the final paper. I will keep my positive score to support your publication.

---

> > > ### Author Response · Authors · 2026-04-03
> > >
> > > Thank you for your reply and support. We will revise the paper accordingly in light of your insightful questions and suggestions.

---

### Official Review · Reviewer_zGEd · 2026-03-13

**Soundness:** 3
**Presentation:** 4
**Significance:** 3
**Originality:** 3
**Overall Recommendation:** 5
**Confidence:** 3

**Summary:**

This paper studies generalization bounds under heavy-tailed gradient noise for non-convex optimization. Specifically, motivated by the fact that most existing results focus on the classical bound variance assumption, the authors derive generalization error bounds based on algorithmic stability under the $p$-th bounded moment noise assumption with $p \in (1,2]$. Furthermore, the authors use the obtained generalization framework to analyze clipped SGD, normalized SGD, normalized SGD with momentum, and normalized SGD with clipping and momentum.

**Compliance With Llm Reviewing Policy:**

Affirmed.

**Final Justification:**

The rebuttal has fully addressed my questions and concerns. I believe that the paper has interesting results and is nicely written with clear explanations. Therefore I have kept my original positive assessment.

**Key Questions For Authors:**

- Why is the assumption $x_0 = 0$ necessary in the theoretical results in Section 5?

**Limitations:**

Yes.

**Strengths And Weaknesses:**

**Strengths**:
- The paper is very clearly written, has a good structure, and clear motivation.
- Most theoretical results are nicely explained with remarks.
- The proof sketch in the main part of the paper explains the introduced truncation argument nicely and it is generally easy to follow.
- The bound in Theorem 4.2, which is one of the main contributions of this work, coincides with previous work in the bounded variance regime ($p=2$).
- The paper establishes stability and population risk bounds specialized to several specific SGD variants and compares the resulting guarantees, which helps clarify whether specific variants can yield improvements in stability and generalization.

**Weaknesses**:
- I did not identify any major weaknesses. However, for completeness and to enable clearer comparisons with mini-batch normalized SGD, it would be beneficial if all SGD variants were also analyzed in the mini-batch setting as well.

---

> ### Author Rebuttal · Authors · 2026-03-30
>
> Thank you for your positive assessment and helpful suggestions.
>
> **Weaknesses:**
>
> > For completeness and to enable clearer comparisons with mini-batch normalized SGD, it would be beneficial if all SGD variants were also analyzed in the mini-batch setting as well.
>
> **A1:** Our analysis for mini-batch normalized SGD is based on Lemma A.3. **This idea also works for other SGD variants in mini-batch setting.**
>
> We take mini-batch version of normalized SGD with momentum as an example, which iterates with
> $$\bar g_t = \frac{1}{B}\sum_{b=1}^B \nabla f(x_t;\xi_{i_t^{(b)}}), \quad
> m_t = \beta m_{t-1} + (1-\beta)\bar g_t, \quad
> x_{t+1} = x_t - \eta \frac{m_t}{\\|m_t\\|}.$$
> Compared with the analysis of standard normalized SGD with momentum in Appendix C.6, we only need to adjust the upper bound on $\mathbb{E}[\\|\zeta_t\\|^p\mid \mathcal{F}_t]$ (line 1191) to
> $$
> \\mathbb\{E\}\\bigl[\\|\\zeta\_t\\|^p \\mid \\mathcal\{F\}\_t\\bigr]=\\frac\{1\}\{B^p\}\\,
> \\mathbb\{E\}\\Bigl[\\Bigl\\|\\sum\_{b=1}^B X\_{t,b}\\Bigr\\|^p \\,\\Bigm|\\, \\mathcal\{F\}\_t\\Bigr]
> \\leq
> \\frac\{2\}\{B^p\}\\sum\_{b=1}^B
> \\mathbb\{E\}\\bigl[\\|X\_{t,b}\\|^p \\mid \\mathcal\{F\}\_t\\bigr]
> \\le 8G^p B^{1-p}.
> $$
> where $X\_{t,b} = \\nabla f\\bigl(x\_t;\\xi\_{i\_t^{(b)}}\\bigr) - \\nabla F\_S(x\_t)$ and $\\zeta\_t = \\frac{1}{B} \\sum\_{b=1}^B X\_{t,b}$.
> Here, the first inequality follows from Lemma A.3.
> The remainder of the analysis for this mini-batch method can directly follow that in lines 1193--1252.
>
> Similarly, Lemma A.3 can also be applied to analyze the mini-batch variants of Clipped SGD and normalized SGD with clipping and momentum.
>
> Thank you for your constructive suggestion. We are happy to add a remark on this discussion in the revision.
>
> **Questions:**
>
> > Why is the assumption $x_0 = 0$ necessary in the theoretical results in Section 5?
>
> **A2:** The assumption of $x\_0=0$ is only for the ease of presentation, which is common in the analysis of stability, e.g., [1, 2].
> Note that **our main results still hold without this assumption**.
> We illustrate this point by taking the example of Clipped SGD.
> If $\\|x\_0\\| \\ne 0$, we can adjust the upper bound of $\\|x\_t\\|$ (the result in line 952) to
> $$
> \\|x\_t\\|
> \\le \\sum\_{s=1}^t \\|x\_s-x\_{s-1}\\| + \\|x\_0\\|
> = \\sum\_{s=0}^{t-1} \\Big\\|\\eta\_s \\cdot \\mathrm{clip}\_{\\gamma}(\\nabla f(x\_s;\\xi\_{i\_s})) \\Big\\| + \\|x\_0\\|
> \\le \\gamma\\sum\_{s=0}^{t-1} \\eta\_s + \\|x\_0\\|.
> $$
> Accordingly, the upper bounds of $\\|A(S)\\|$ and $\\|A(S')\\|$ (the result in line 957) become
> $$
> \\|A(S)\\|\\le \\gamma\\sum\_{t=0}^{T-1}\\eta\_t + \\|x\_0\\|
> \\quad \\text{and} \\quad
> \\|A(S')\\|\\le \\gamma\\sum\_{t=0}^{T-1}\\eta\_t + \\|x\_0\\|.
> $$
> Following the above results and the derivations in lines 960--980 by taking $\\eta\_t=\\eta$, we can achieve the following stability bound for clipped SGD:
> $$
> \\mathbb{E}\_{A}\\bigl[\\|A(S)-A(S')\\|^2\\bigr]
> \\le 8\\gamma^2\\Bigl(\\sum\_{t=0}^{T-1}\\eta\_t\\Bigr)^2\\frac{T}{n} +  \\frac{8T\\|x\_0\\|^2}{n}
> = \\frac{8\\gamma^2\\eta^2 T^3}{n} + \\frac{8 T \\|x\_0\\|^2}{n}.
> $$
> The above upper bound is dominated by the term $8\\gamma^2\\eta^2 T^3/n$, which does not depend on $||x\_0||$.
> Hence, assuming $x\_0=0$ does not affect our main results.
>
> We will point out in the revision that our main results still hold without the assumption $x\_0=0$.
>
> ---
> References
>
> [1] Yunwen Lei. Stability and generalization of stochastic optimization with nonconvex and nonsmooth problems. In COLT 2023.
>
> [2] Matan Schliserman and Tomer Koren. Stability vs implicit bias of gradient methods on separable data. In COLT 2022.

---

> > ### Author Rebuttal · Reviewer_zGEd · 2026-04-02
> >
> > I thank the authors for their detailed response. I hope that the revised version of the paper can incorporate the proposed suggestions to further improve clarity. As my recommendation is already for acceptance, I will keep my score.

---

> > > ### Author Response · Authors · 2026-04-03
> > >
> > > Thank you for your reply and helpful suggestions. We will revise the paper accordingly and further improve the final version.

---

### Decision · Program_Chairs · 2026-04-30

**Decision:**

Accept (regular)

**Comment:**

This paper studies the stability and generalization of nonconvex optimization under heavy-tailed gradient noise with $p$-th bounded central moment assumptions for $p\in (1,2]$. It proposes a smart truncation-and-tail decomposition technique to derive generalization error bounds based on algorithmic stability, covering multiple SGD variants like clipped SGD, normalized SGD with momentum and their mini-batch versions. The paper received four detailed reviews, all acknowledging its clear writing, solid theory, significant novelty in bridging stability and heavy-tailed regimes, and consistent results with prior work when $p=2$. Minor concerns about mini-batch analysis, initial assumption $x_0=0$, bound sharpness and literature updates were fully resolved in the rebuttal.

Overall, the submission makes novel and impactful contributions to the generalization theory of nonconvex stochastic optimization under heavy-tailed noise. The meta reviewer is pleased to recommend the paper for acceptance.